# Optimal Transport-based Labor-free Text Prompt Modeling for Sketch Re-identification

**Rui Li,*** **Tingting Ren,*** **Jie Wen,†** **Jinxing Li†**
School of Computer Science and Technology, Harbin Institute of Technology, Shenzhen
lrhit@stu.hit.edu.cn, {rentt0410,lijinxing158}@gmail.com, jiewen_pr@126.com

## Abstract

Sketch Re-identification (Sketch Re-ID), which aims to retrieve target person from an image gallery based on a sketch query, is crucial for criminal investigation, law enforcement, and missing person searches. Existing methods aim to alleviate the modality gap by employing semantic metrics constraints or auxiliary modal guidance. However, they incur expensive labor costs and inevitably omit fine-grained modality-consistent information due to the abstraction of sketches. To address this issue, this paper proposes a novel *Optimal Transport-based Labor-free Text Prompt Modeling* (OLTM) network, which hierarchically extracts coarse- and fine-grained similarity representations guided by textual semantic information without any additional annotations. Specifically, multiple target attributes are flexibly obtained by a pre-trained visual question answering (VQA) model. Subsequently, a text prompt reasoning module employs learnable prompt strategy and optimal transport algorithm to extract discriminative global and local text representations, which serve as a bridge for hierarchical and multi-granularity modal alignment between sketch and image modalities. Additionally, instead of measuring the similarity of two samples by only computing their distance, a novel triplet assignment loss is further proposed, in which the whole data distribution also contributes to optimizing the inter/intra-class distances. Extensive experiments conducted on two public benchmarks consistently demonstrate the robustness and superiority of our OLTM over state-of-the-art methods.

## 1 Introduction

With the growing need for urban public safety, traditional person re-identification (Re-ID) methods [1, 2, 3] are gradually becoming inadequate for criminal investigations and missing person tracking, as the individuals of interest may not have been captured by surveillance cameras. To bolster social security management and combat criminal activities, sketch person re-identification (Sketch Re-ID), which utilizes eyewitness clues to draw professional sketches as queries and match target images in a photo gallery database, has received widespread attention from researchers and scholars [4, 5, 6], as shown in Fig. 1. Nonetheless, due to the considerable disparity in modal heterogeneity resulting from the varied sketch styles of different artists and the

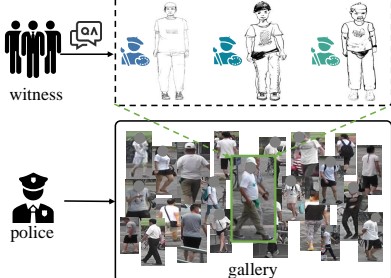

Figure 1: The illustration of sketch Re-ID. Different artists create sketches based on clues provided by witness to assist the police in identifying targets.

---

*Equal Contribution.
†Corresponding Author.

38th Conference on Neural Information Processing Systems (NeurIPS 2024).

diverse postures of real pedestrians in monitoring, sketch Re-ID remains highly challenging and necessitates further exploration and investigation.

Giving the high generalization and inherent abstraction of person characteristics in sketches, the pedestrian features depicted in a single sketch (such as clothing and gender) may match multiple similar real images, as illustrated in Fig. 1. A viable solution is to leverage the inter-modality interaction within the feature space to achieve hard alignment. Such methods typically employ loss constraints to directly map different modalities into a generic latent space [4, 7, 8, 9]. However, this hard alignment manner may not fully capture the complex dependencies and correlations that exist within and across modalities. To compensate for the lack of details in the above manner, another branch is to introduce an intermediate modality to bridge two source modalities. For instance, [10, 11] generate simulated sketches through adversarial learning, but the generated sketches are inevitably corrupted with noise due to limited generation performance. Additionally, [7, 12] construct benchmarks that contain textual information to alleviate modal gap; [6] improves inference efficiency by introducing text only during the training process. Despite the fact that these additional texts do contribute to mitigating the modal gap, they are all manually labeled, requiring significant human labor in real-world applications. Moreover, existing text-guided methods [6, 7] only focus on global text embeddings as masks, neglecting finer and richer local features. Therefore, this paper aims to address two key challenges: **i) developing sufficient textual information as a transition mechanism without incurring additional costs**, and **ii) further exploring fine-grained discriminative information for multi-granularity interaction.**

To address the above issues, we propose the Optimal Transport-based Labor-free Text Prompt Modeling (OLTM) framework, which implicitly incorporates text semantic information during training, facilitating hierarchical and multi-granularity modal alignment. In particular, OLTM is composed of three main components: i) text prompt reasoning (TPR); ii) text-injected coarse-grained alignment module (TCA); iii) consensus-guided fine-grained interaction module (CFI). On the one hand, to introduce text sequences without additional manual annotation, we dynamically transfer pre-trained language-visual knowledge into the downstream task. Specifically, TPR first generates multi-dimensional person attributes based on real images with a pre-trained visual question answering (VQA) model. Then, these attributes are inserted as fixed parts into learnable prompts to obtain the textual embedding representations. TCA integrates global parts of the embeddings to achieve the coarse-grained alignment across modalities. On the other hand, to explore fine-grained information, we employ optimal transport theory to enhance deep-level interaction. Concretely, TPR formulates the mapping from local parts of the textual embeddings to more discriminative feature representations, i.e., consensus, as an optimal transport problem. Subsequently, guided by consensus, CFI selectively focuses on key details, extracting fine-grained conceptual representations for sketch-ID. In addition, due to the significant heterogeneity gap between modalities, using Euclidean distance as a sole metric to measure feature similarity is inadequate. Thus, we propose a triplet assignment loss to optimize feature distance measurement and improve model performance. Extensive experiments are conducted on two challenging benchmarks, which demonstrate the favorable comparison of OLTM with other state-of-the-art methods.

The core contributions of this work are summarized as follows: (1) This paper proposes a novel optimal transport-based labor-free text prompt modeling framework for sketch Re-ID. To our best knowledge, this is *the first attempt* to apply VQA-generated text responses as a means to achieve modal alignment in sketch Re-ID *without any additional annotations*. (2) A novel text prompt reasoning module is deployed to dynamically extract global textual embeddings and discriminative fine-grained consensus, which guide the hierarchical multi-granularity alignment module in injecting semantic knowledge into the modeling process. (3) A new triplet assignment loss is proposed, which optimizes inter-/intra-class distance by considering overall data distribution information.

## 2   Related Work

**Sketch Re-identification**   As an important part of public safety guarantee, sketch Re-ID is a novel and challenging task that aims to match a person image with given professional sketches. Existing sketch Re-ID methods could be roughly classified into two groups according to their interaction modes, i.e., hard alignment methods [4, 8, 9] and soft alignment methods [12, 10, 6]. The former try to learn modality embeddings in a common latent space by employing some modality interaction operations or semantic metrics. Pang *et al*. [4] pioneered a sketch-photo benchmark and introduced

cross-domain adversarial learning to narrow the feature gap. Zhang *et al*. [9] proposed an advanced cross-modal learning mechanism for handling non-corresponding information between modalities. However, due to significant differences between modalities, this direct alignment paradigm inevitably loses fine-grained modality-specific cues [13]. Hence, some of the latter methods investigate gentler alignment techniques through transitional modality. For example, Chen *et al*. [10] designed a dynamic updatable auxiliary sketch modality to increases the diversity of training samples; Zhai *et al*. [12] introduced a multi-modal Re-ID task by combining text and sketch as query for retrieval, exploiting their complementary advantages. Obviously, auxiliary modalities lacking detailed information may introduce noise, while data annotation is a labor-intensive task. In this paper, we *first attempt* to use text attributes generated by a reasonable VQA model as guidance for achieving multi-granularity alignment across modalities in sketch Re-ID.

**Optimal Transport** For optimizing the moving cost between distributions, Optimal Transport (OT) was first proposed by Kantorovich [14], which has shown significant potential in machine learning and computer vision, e.g., domain adaptation [15, 16, 17], learning with noisy labels [18, 19], and feature matching [20, 21]. Zhang *et al*. [22] incorporated OT into the re-ranking phase of image retrieval, significantly improving accuracy and efficiency. Similarly, Sergio *et al*. [23] first applied OT in visual place recognition and introduced a novel local feature aggregation method. In semi-supervised person Re-ID, OT often achieves the mapping between pseudo labels and classes as a classifier [24, 25]. In addition, Ling *et al*. [26] designed a assignment strategy for alleviating the intra-identity variations; Wasserstein distance was used to rectify the original global distance between samples and provides aligned distance estimation for local features [27]. Considering the enormous potential of OT in feature aggregation and distribution mapping, our study adopts OT to assist fine-grained alignment between modalities and guide the model in extracting the overall sample distribution pattern.

**Prompt Learning** Prompt learning initially garnered widespread attention and extensive research in natural language processing [28, 29, 30], which has gradually demonstrated significant potential in vision-language (V-L) models [31, 32, 33] and pure vision models [34, 35, 36]. Prompt learning provides a flexible way to adapt pre-trained models to downstream tasks by training only additional parameters. This enables prompts to capture task-specific information while guiding the fixed model's performance [37, 38]. In sketch-based image retrieval, [39] innovatively learns a unified prompt for different branches in CLIP's [40] visual encoding layer, fully exploiting CLIP's zero-shot learning potential. In text-to-image person Re-ID, [41] introduces a multi-prompt strategy to integrate text prompts from various sources for fine-grained interaction. Furthermore, Li et al. [42] first attempt to conduct in-depth research on zero-shot multi-modal ReID through a large foundational model. In this paper, we delve into the significant application of prompt learning in sketch Re-ID, innovatively generating global text representations by integrating fixed and learnable prompts, and utilizing OT to reason consensus for effectively guiding detailed interactions across modalities.

## 3 Preliminaries

### 3.1 Problem Statement

To ensure clarity, we represent the gallery containing $m$ images $I$ as $\mathcal{G} = \{I_i, y_i\}_{i=1}^m$ and the query set containing $n$ sketches $S$ as $\mathcal{S} = \{S_j, y_j\}_{j=1}^n$, where $y \in \{1, \ldots, C\}$ are the identity labels for $C$ distinct pedestrian entities. Notably, each entity may include multiple images and sketches. The goal of Sketch Re-ID is to retrieve pedestrian images from the gallery $\mathcal{G}$ that match one or multiple given sketch. Like [6], there exist two types of query methods: single sketch query and multiple sketches query. This section will use single sketch query as an example, and the same applies to multiple sketches query. Formally, we define a matching function $\mathcal{M} : \mathcal{S} \times \mathcal{G} \to \mathbb{R}^{n \times m}$ that assigns a similarity score to each pair $(I_i, S_j)$. The objective is to learn a function $\mathcal{M}$ such that for any sketch $S_j$ and image $I_i$,

$$\mathcal{M}(I_i, S_j) > \mathcal{M}(I_k, S_j) \quad if \quad y_i = y_j \ and \ y_k \neq y_j, \tag{1}$$

where $I_i$ and $I_k$ are images from the gallery set, and $y_i$ and $y_k$ are their respective identity labels.

### 3.2 Optimal Transport

Optimal Transport is a mathematical theory that focuses on finding an efficient solution between two probability distributions, minimizing the cost of transporting one distribution into another. We briefly review the theoretical derivation of optimal transport. Let $\Gamma_r := \{\boldsymbol{x} \in \mathbb{R}_+^r | \boldsymbol{x}^\top \mathbf{1}_r = 1\}$ represents

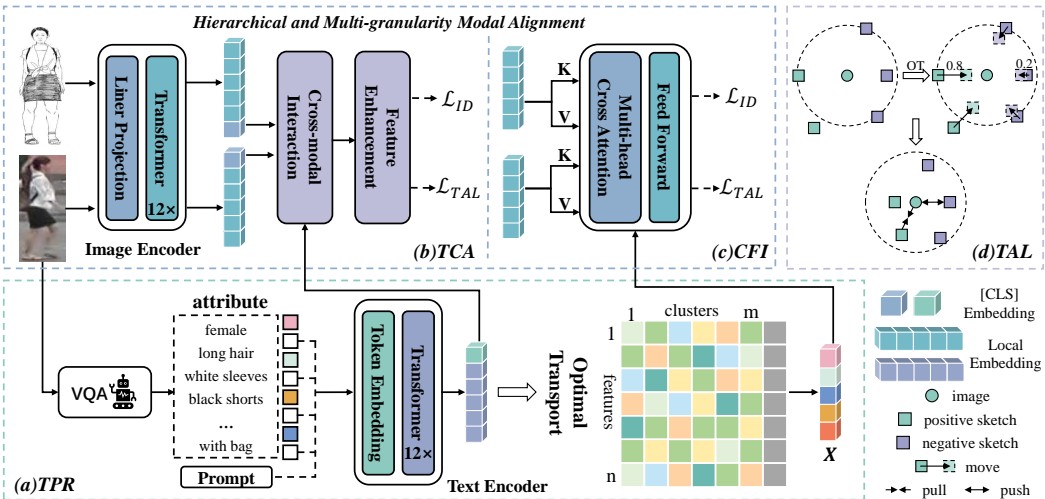

Figure 2: Overview of our proposed OLTM network. Our model includes four main parts, i.e., text prompt reasoning (TPR), text-injected coarse-grained alignment Module (TCA), consensus-guided fine-grained interaction module (CFI) and triplet assignment loss (TAL). Specifically, TPR flexibly generates target characteristics through VQA, and combines prompt learning and optimal transport to reason text global embedding and local consensus. TCA and CFI extract modality-specific representations from image and sketch modalities to achieve hierarchical and multi-granularity alignment. Finally, TAL is designed to optimize distance measurement between samples and improve the model's capacity to capture local relationships.

the probability simplex, where $\mathbf{1}_r$ is the $r$-dimensional vector of ones. Given two probability simplex vectors $\boldsymbol{\alpha} \in \Gamma_m$ and $\boldsymbol{\beta} \in \Gamma_n$ and a cost matrix $\boldsymbol{C} \in \mathbb{R}^{m \times n}$, the objective of OT is to seek the optimal transport plan $\boldsymbol{P}^*$ mapping $\boldsymbol{\alpha}$ to $\boldsymbol{\beta}$ at the minimum cost:

$$d_{\boldsymbol{C}}(\boldsymbol{\alpha}, \boldsymbol{\beta}) = \min_{\boldsymbol{P} \in U(\boldsymbol{\alpha}, \boldsymbol{\beta})} \langle \boldsymbol{C}, \boldsymbol{P} \rangle,$$
$$\boldsymbol{U}(\boldsymbol{\alpha}, \boldsymbol{\beta}) = \left\{ \boldsymbol{P} \in \mathbb{R}_+^{m \times n} \mid \boldsymbol{P} \mathbf{1}_n = \boldsymbol{\alpha}, \boldsymbol{P}^\top \mathbf{1}_m = \boldsymbol{\beta} \right\},$$

(2)

where $\boldsymbol{U}(\boldsymbol{\alpha}, \boldsymbol{\beta})$ denotes the transport polytope of $\boldsymbol{\alpha}$ and $\boldsymbol{\beta}$, i.e., the solution space of $\boldsymbol{P}$. The above problem is to find optimal solution $\boldsymbol{P}^*$ in a set of all possible joint probabilities of $(X, Y)$, where $X$ and $Y$ represent random variables with marginal distribution $\boldsymbol{\alpha}$ and $\boldsymbol{\beta}$.

Eq. 8 indicates that OT is a linear programming problem which is theoretically solvable in polynomial time, but its complexity becomes prohibitively high as the feature dimension increases [43]. To this end, Sinkhorn algorithm [44] adopts an iterative strategy to obtain the optimal solution $\boldsymbol{P}^* = Diag(\boldsymbol{u}) \boldsymbol{K} Diag(\boldsymbol{v})$ with near-square complexity [45]. $\boldsymbol{u}$ and $\boldsymbol{v}$ can be solved through alternately iterating the following two equations: $\boldsymbol{u}^{(z)} = \boldsymbol{\alpha}/(\boldsymbol{K}\boldsymbol{v}^{(z-1)})$ and $\boldsymbol{v}^{(z)} = \boldsymbol{\beta}/(\boldsymbol{K}^\top \boldsymbol{u}^{(z)})$, where $\boldsymbol{K} = exp(\boldsymbol{C}/\epsilon)$, $\epsilon$ is the regularization coefficient and $z$ is the iterations (cf. **Appendix**). Since this method integrates the importance of all features when solving the optimal solution, it can analyze the overall data distribution.

## 4 The Proposed Method

### 4.1 Overall Architecture

Fig. 2 provides an overview of the OLTM architecture. The image and text encoders discussed in this paper are based on CLIP [40], and any language-visual model utilizing a Transformer architecture may also be employed. Notably, the image encoders utilized for both images and sketches employ shared weights to ensure the mapping of features into a unified semantic space. For an input RGB image, we obtain embeddings $\boldsymbol{R} = \{\boldsymbol{R}_{\text{cls}}, \boldsymbol{r}_1, \ldots, \boldsymbol{r}_p\} \in \mathbb{R}^{(p+1) \times d}$ through the image encoder, where $p$ is the number of non-overlapping patches, $\boldsymbol{R}_{\text{cls}}$ and $\boldsymbol{R}_{\text{local}} = \{\boldsymbol{r}_i\}_{i=1}^p$ represent $d$-dimensional global and local features, respectively. Similarly, the embeddings of a sketch can be represented as $\boldsymbol{S} = \{\boldsymbol{S}_{cls}, \boldsymbol{S}_{local}\} = \{\boldsymbol{S}_{\text{cls}}, \boldsymbol{s}_1, \ldots, \boldsymbol{s}_p\} \in \mathbb{R}^{(p+1) \times d}$. Firstly, to provide reasonable text semantic

guidance, Text Prompt Reasoning (TPR) generates attribute descriptions about pedestrians based on the RGB image, and obtains the textual embeddings $T = \{T_{eos}, T_{local}\}$ through prompt learning. Then, TPR extracts the fine-grained consensus $X$ from $T_{local}$ through clustering. Subsequently, Text-injected Coarse-grained Alignment (TCA) module embeds global contextual information $T_{eos}$ into visual features $R_{cls}$ and $S_{cls}$. Meanwhile, Consensus-guided Fine-grained Interaction (CFI) module utilizes $X$ to address fine-grained semantic misalignment between $R_{local}$ and $S_{local}$. Additionally, due to the significant differences between sketches and RGB images, Euclidean distance between independent samples may ignore the influence of overall sample distribution. Thus, we introduce a more comprehensive distance measurement method and propose triplet assignment loss $\mathcal{L}_{tal}$. During training, all these modules will be jointly optimized through identity loss [46] $\mathcal{L}_{id}$ and $\mathcal{L}_{tal}$: $\mathcal{L}_{OLTM} = \mathcal{L}_{id} + \eta\mathcal{L}_{tal}$, where $\eta$ is a scaling factor. During inference, only $R_{cls}$ and $S_{cls}$ are used to match queries for practical application requirements.

## 4.2 Text Prompt Reasoning

Significant image differences and inherent abstract nature, cause semantic misalignment during knowledge acquisition, severely impacting model's reasoning and generalization capabilities. To address this issue, TPR introduces intermediate modality to guide alignment between modalities without additional costs. Moreover, TPR employs a dynamic consensus acquisition strategy to enhance the discriminative power of local text features.

**Text Attribute Generation** Sketches, unlike conventional Re-ID tasks, are vulnerable to subjective emotions and drawing skills of artists, leading to a lack of detailed information crucial for model learning. The text's objectivity and flexibility prompt the model to focus more on semantic contextual information during knowledge acquisition. However, directly generating a comprehensive textual description of pedestrian images inevitably introduces irrelevant noise, thereby reducing model performance. Therefore, we retain the advanced modeling capabilities of large-scale language-visual models for images as possible. Specifically, for a given RGB image, TPR utilizes a pre-trained visual question answering model to address $k$ specific details (cf. **Appendix**) and acquire corresponding descriptions for the target: $att = \{att_1, att_2, \ldots, att_k\}$. Importantly, this process introduces textual detail guidance during model training but excludes text-related components during inference.

**Learnable Prompt Strategy** Inspired by [47], we combine the learnable prompt with original text attributes, without incurring additional expert knowledge compared to the handcrafted prompt. Concretely, TPR initially transform these attributes into tokens through CLIP tokenizer, i.e., $a = Tokenizer(att)$. Then, $l$ learnable prompts $\{p_1, p_2, \ldots, p_l\}$ are embedded into these fixed attributes tokens, forming the textual description: $q = \{p_1, a_1, p_2, a_2, \ldots, p_l, a_k\}$. This integration introduces a dynamic knowledge learning mechanism that reduces noise introduction compared to handcrafted prompts, while enhancing the flexibility of modal interaction and transferability of text embeddings. Subsequently, the whole token $q$ is fed into a frozen text encoder to generate text embeddings $T = \{T_{sos}, t_1, t_2, \ldots, t_n, T_{eos}\} \in \mathbb{R}^{(n+2) \times d}$, where $T_{sos}$ and $T_{eos}$ denote the [SOS] and [EOS] token, $n$ is the number of $d$-dimensional word tokens. Based on widely-used token selection, $T_{eos}$ serves as the global feature, while $T_{local} = \{t_j\}_{j=1}^{n}$ represents a sequence of basic local tokens.

**Dynamic Consensus Acquisition** To more effectively address fine-grained semantic variations (e.g., hats, shoes) across modalities, we explore methods to filter out non-informative features for enhancing the representational capacity of text embedding for detailed information. Therefore, based on local textual feature $T_{local}$, we employ metric learning to draft a dynamic consensus acquisition strategy for capturing the discriminative prototypical representations $X$.

To begin, in order to adaptively learn related-task knowledge, local text representations $T_{local} \in \mathbb{R}^{n \times d}$ are fed into consensus multi-layer perceptron (ConMLP) blocks to achieve feature enhancement. Then, a prototypical descriptor is formed by assigning all enhanced features to a set of atoms. The cost matrix $C \in \mathbb{R}_+^{n \times m}$ can be calculated for assignment, where the $(i, j)$-th element $C_{i,j}$ represents the cost of assigning a feature to an atom. In other words, $C$ evaluates the affinity between the enhanced features and the prototypical atoms. Concretely, to introduce certain priors as guidance [23], the enhanced features are used to learn the cost matrix through two fully connected layers initialized randomly. In addition, some irrelevant information in textual features, such as those representing association, may disrupt the model's ability to learn the target details. Inspired by the solutions used in graph matching and key-point matching [20, 23], we set a learnable "bin" in $C$. Due to the differences between prior distributions, non-informative features can be assigned to it. Specifically,

we extend the cost matrix from $\boldsymbol{C}$ to $\bar{\boldsymbol{C}} = [\boldsymbol{C}, \bar{\boldsymbol{c}}] \in \mathbb{R}_+^{n \times (m+1)}$, and $\bar{\boldsymbol{c}} = w\mathbf{1}_n$, where $w$ is a learnable parameter and $\mathbf{1}_n = [1, \ldots, 1]^\top \in \mathbb{R}^n$ represents $n$-dimensional vector of ones. Following Sec. 3.2, we consider that assignment where the enhanced features' mass, $\boldsymbol{\alpha} = \mathbf{1}_n$, should be assigned to the atoms or the "bin", $\boldsymbol{\beta} = [\mathbf{1}_m^\top, n - m]^\top$, is an optimal transport problem:

$$d_{\bar{\boldsymbol{C}}}(\boldsymbol{\alpha}, \boldsymbol{\beta}) = \min_{\boldsymbol{P} \in \boldsymbol{U}(\boldsymbol{\alpha}, \boldsymbol{\beta})} \langle \bar{\boldsymbol{C}}, \boldsymbol{P} \rangle,$$
$$\boldsymbol{U}(\boldsymbol{\alpha}, \boldsymbol{\beta}) = \{\boldsymbol{P} \in \mathbb{R}_+^{n \times (m+1)} | \boldsymbol{P}\mathbf{1}_{m+1} = \boldsymbol{\alpha}, \boldsymbol{P}^\top \mathbf{1}_n = \boldsymbol{\beta}\}. \tag{3}$$

The optimal assignment plan can be solved by Sinkhorn algorithm (cf. **Appendix**) through iterative strategy. For better descriptors quality, the extra "bin" is discarded to obtain the optimal assignment $\boldsymbol{P} \in \mathbb{R}_+^{n \times m}$. Finally, the augmented comprehensive representation, i.e., consensus $\boldsymbol{X} \in \mathbb{R}^{m \times d}$, is obtained by aggregating enhanced textual features and optimal transport plan: $\boldsymbol{X} = \boldsymbol{P}^\top \boldsymbol{T}_{local}$.

### 4.3 Hierarchical and Multi-granularity Modal Alignment

TPR develops a robust transitional means to facilitate reasonable and efficient modal alignment. In order to leverage effectively the multi-granularity textual information, TCA utilizes global embedding to achieve coarse-grained modal interaction, while CFI optimizes fine-grained alignment based on enhanced comprehensive representation.

**Text-injected Coarse-grained Alignment Module** TCA utilizes transformer architecture, known for its efficiency in modeling long-distance dependencies, to capture global information, as shown in Fig. 2. Furthermore, we implement a cross-attention mechanism using global text embedding to emphasize contextual information injection at the beginning. For an input pair $(\boldsymbol{R}, \boldsymbol{S})$, cross-attention mechanism computes the *query* using text global representation $\boldsymbol{T}_{eos}$ and derives the *key* and *value* from modality-specific global features $\boldsymbol{R}_{cls}$ and $\boldsymbol{S}_{cls}$ :

$$\boldsymbol{Q}_{R/S} = \boldsymbol{T}_{eos} \cdot \boldsymbol{w}^Q, \boldsymbol{K}_{R/S} = (\boldsymbol{R}/\boldsymbol{S})_{cls} \cdot \boldsymbol{w}^K, \boldsymbol{V}_{R/S} = (\boldsymbol{R}/\boldsymbol{S})_{cls} \cdot \boldsymbol{w}^V,$$
$$CA(\boldsymbol{R}/\boldsymbol{S}, \boldsymbol{T}_{eos}) = Attention(\boldsymbol{Q}_{R/S}, \boldsymbol{K}_{R/S}, \boldsymbol{V}_{R/S}), \tag{4}$$

where $\boldsymbol{R}/\boldsymbol{S}$ signifies identical operations across both modalities; $\boldsymbol{w}^Q$, $\boldsymbol{w}^K$ and $\boldsymbol{w}^V$ denote shared learnable parameters, while $\boldsymbol{Q}_{R/S}$, $\boldsymbol{K}_{R/S}$ and $\boldsymbol{V}_{R/S}$ represent *query*, *key* and *value* for either the RGB or sketch modality, respectively. After obtaining the fusing features that contain textual knowledge, TCA introduces the standard transformer blocks to refine modality-specific features. Finally, we can acquire the final global concept representations $\boldsymbol{R}_{cls}^{'}$ and $\boldsymbol{S}_{cls}^{'}$ for sketch Re-ID.

**Consensus-guided Fine-grained Interaction Module** Due to the inherent complexity of sketches and RGB images and the potential for semantic misalignment during learning, capturing detail variations in modalities is crucial. Fortunately, the fine-grained consensus $\boldsymbol{X}$ provided by TPR, which contains rich detail information, offers a key solution to this problem. CFI adopts a transformer structure based on multi-head cross attention, and it converts the original local feature $\boldsymbol{R}_{local}$ and $\boldsymbol{S}_{local}$ to more discriminative representations through for robust Re-ID:

$$\hat{\boldsymbol{Q}}_{R/S} = \boldsymbol{X} \cdot \boldsymbol{w}^{\hat{Q}}, \hat{\boldsymbol{K}}_{R/S} = (\boldsymbol{R}/\boldsymbol{S})_{local} \cdot \boldsymbol{w}^{\hat{K}}, \hat{\boldsymbol{V}}_{R/S} = (\boldsymbol{R}/\boldsymbol{S})_{local} \cdot \boldsymbol{w}^{\hat{V}},$$
$$Head_h^{R/S} = Attention(\hat{\boldsymbol{Q}}_{R/S}, \hat{\boldsymbol{K}}_{R/S}, \hat{\boldsymbol{V}}_{R/S}), \tag{5}$$
$$MH(\boldsymbol{R}/\boldsymbol{S}, \boldsymbol{X}) = Concat(Head_1^{R/S}, \ldots, Head_H^{R/S}),$$

where $\boldsymbol{w}^{\hat{Q}}$, $\boldsymbol{w}^{\hat{K}}$ and $\boldsymbol{w}^{\hat{V}}$ denote the parameters of project layers of $h$-th head for both modalities. As a result, the text detail information about pedestrian characteristics can assist our model to address the problem caused by the diversity and uncertainty of sketches and RGB images. Eventually, the local concept representations $\boldsymbol{R}_{local}^{'}$ and $\boldsymbol{S}_{local}^{'}$ can be obtained.

### 4.4 Triplet Assignment Loss

The triple loss, a commonly-used matching loss in cross-modal learning, performs well in performance through adjusting the distance of hardest negatives in scenarios like image-text matching [12, 48] and video-text retrieval [49]. However, this strategy independently trains each semantic part with equal contribution, disregarding the overall data distribution impact when multiple samples from

Table 1: Comparison with state-of-the-art methods on Market-Sketch-1K dataset. Both training and testing set uses all sketches. 'S' and 'M' represent single-query and multi-query, respectively. 'Backbone' refers to network structure used by each method, mainly including: ResNet50 [50] and CLIP [40]. **Bold** values represent the optimal results.

| Methods | Query | Backbone | Reference | mAP | Rank@1 | Rank@5 | Rank@10 |
|---------|-------|----------|-----------|-----|--------|--------|---------|
| DDAG [51] | S | ResNet50 | ECCV'2020 | 12.13 | 11.22 | 25.40 | 35.02 |
| CM-NAS [52] | S | ResNet50 | ICCV'2021 | 0.82 | 0.70 | 2.00 | 3.90 |
| CAJ [53] | S | ResNet50 | ICCV'2021 | 2.38 | 1.48 | 3.97 | 7.34 |
| MMN [54] | S | ResNet50 | MM'2021 | 10.41 | 9.32 | 21.98 | 29.58 |
| DART [55] | S | ResNet50 | CVPR'2022 | 7.77 | 6.58 | 16.75 | 23.42 |
| DCLNet [56] | S | ResNet50 | MM'2022 | 13.45 | 12.24 | 29.20 | 39.5 |
| DSCNet [57] | S | ResNet50 | TIFS'2022 | 14.73 | 13.84 | 30.55 | 40.34 |
| DEEN [58] | S | ResNet50 | CVPR'2023 | 12.62 | 12.11 | 25.44 | 30.94 |
| BDG [6] | S | ResNet50 | MM'2023 | 19.61 | 18.10 | 38.95 | 50.75 |
| | M | | | 24.45 | 24.70 | 50.40 | 63.45 |
| UNIReID [7] | S | CLIP | CVPR'2023 | 34.97 | 31.52 | 57.17 | 70.46 |
| | M | | | 55.18 | 56.63 | 82.33 | 91.97 |
| OLTM (Ours) | S | CLIP | - | **38.35** | **36.75** | **63.88** | **74.05** |
| | M | | | **62.55** | **69.48** | **90.36** | **95.18** |

different modalities exhibit slight differences. This oversight may lead to inaccurate estimation of sample distances and potentially result in sub-optimal local minima [27]. To this end, we propose a new triplet assignment loss (TAL) to establish a more rational measure for evaluating the proximity of local features.

For an input positive pair $(\boldsymbol{R}_i, \boldsymbol{S}_i)$ in a mini-batch $x$, the feature representations obtained through model inference are $\boldsymbol{R}_i^{'}$ and $\boldsymbol{S}_i^{'}$. If we treat the feature sets of all samples for each of two modalities in $x$ as two discrete distributions, their alignment can be considered an optimal transport problem. The cost matrix $\hat{\boldsymbol{C}}$ is derived from pairwise feature similarities: $\hat{\boldsymbol{C}}_{i,j} = [(\boldsymbol{R}_i^{'})^{\top} \boldsymbol{S}_j^{'}]_+$. We aim to acquire the optimal transport matrix $\boldsymbol{P}^*$ with the least amount of cost, where $\boldsymbol{P}_{i,j}^*$ represents the assignment weight of $(\boldsymbol{R}_i, \boldsymbol{S}_j)$ obtained after balancing the overall distribution. TAL can be represented based on triplet loss as the weighted sum of the original distance and the optimal assignment distance, dynamically updated at a certain rate $\gamma$:

$$\mathcal{L}_{tal}(\boldsymbol{R}_i, \boldsymbol{S}_i) = [m - D(\boldsymbol{R}_i, \boldsymbol{S}_i) + D(\boldsymbol{R}_i, \hat{\boldsymbol{S}}_h)]_+ + [m - D(\boldsymbol{R}_i, \boldsymbol{S}_i) + D(\hat{\boldsymbol{R}}_h, \boldsymbol{S}_i)]_+,$$
$$D(\boldsymbol{R}_i, \boldsymbol{S}_i) = \gamma E(\boldsymbol{R}_i, \boldsymbol{S}_i) + (1 - \gamma)(1 - \boldsymbol{P}_{i,i}^*) E(\boldsymbol{R}_i, \boldsymbol{S}_i) \tag{6}$$

where $[x]_+ = \max(x, 0)$, $\hat{\boldsymbol{R}}_h = argmax_{R_j \neq R_i} D(\boldsymbol{R}_j, \boldsymbol{S}_i)$ and $\hat{\boldsymbol{S}}_h = argmax_{S_j \neq S_i} D(\boldsymbol{R}_i, \boldsymbol{S}_j)$ are the most similar negatives in $x$ for $(\boldsymbol{R}_i, \boldsymbol{S}_i)$, and $E(\boldsymbol{R}_i, \boldsymbol{S}_i) = \|\boldsymbol{R}_i^{'} - \boldsymbol{S}_i^{'}\|_2$ denotes the Euclidean distance between feature representations.

## 5 Experiment

### 5.1 Experiment Setup

**Datasets** Two publicly available benchmark datasets, namely PKU-Sketch [4] and Market-Sketch-1K [6], are utilized for performance evaluation. Both of them are sketched and annotated by professional artists. **PKU-Sketch** is the first publicly Sketch Re-ID dataset, containing data for 200 pedestrians, with each individual being represented through one sketch and two photos. In accordance with the setting of [4], we randomly select 150 identities for training and 50 for testing, and final results are derived from the average of 10 experimental runs. **Market-Sketch-1K** is a large-scale dataset derived from the Market-1501 [1], which is created by six artists based on descriptions, featuring multiple perspectives and artistic styles. The training set consists of 2,332 sketches and 12,936 photos, while the testing set comprises 2,375 sketches and 19,732 photos. Following the experimental setup in [6], our method will be evaluated in three settings: single-query, multi-query, and cross-style retrieval.

Table 2: Comparison with state-of-the-art methods on PKU-Sketch dataset. 'Backbone' includes GoogleNet [62], VGG-16 [63], ResNet50, ViT [64], and CLIP. '-' denotes the unavailable results. '†' indicates that we reproduce UNIReID results following our training configuration.

| Methods | Backbone | Reference | mAP | Rank@1 | Rank@5 | Rank@10 |
|---|---|---|---|---|---|---|
| TripleSN [65] | - | CVPR'2016 | - | 9.0 | 26.8 | 42.2 |
| GNSiamese [66] | GoogleNet | TOG'2016 | - | 28.9 | 54.0 | 62.4 |
| AFLNet [4] | GoogleNet | MM'2018 | - | 34.0 | 56.3 | 72.5 |
| LMDI [8] | VGG-16 | Neuro'2020 | - | 49.0 | 70.4 | 80.2 |
| SKetchTrans [10] | ViT | MM'2022 | - | 84.6 | 94.8 | 98.2 |
| CCSC [9] | ViT | MM'2022 | 83.7 | 86.0 | 98.0 | **100.0** |
| SKetchTrans+ [5] | ViT | PAMI'2023 | - | 85.8 | 96.0 | 99.0 |
| UNIReID† [7] | CLIP | CVPR'2023 | 88.7 | 92.4 | 98.0 | 99.6 |
| DALNet [11] | ResNet50 | AAAI'2024 | 86.2 | 90.0 | 98.6 | **100.0** |
| OLTM (Ours) | CLIP | - | **91.4** | **94.0** | **99.4** | **100.0** |

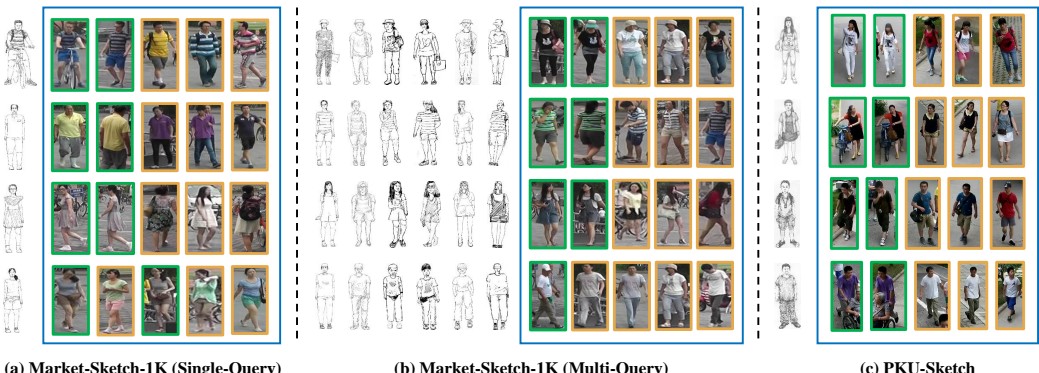

**(a) Market-Sketch-1K (Single-Query)**   **(b) Market-Sketch-1K (Multi-Query)**   **(c) PKU-Sketch**

Figure 3: The Rank-5 retrieval results on two datasets. For the Market-Sketch-1K dataset, both single-query and multi-query scenarios are presented. **Green** border indicates correctly retrieved target pedestrians, while **yellow** border indicates incorrectly matched pedestrians.

**Evaluation Metrics** In line with [4, 9, 6, 59], we use Rank-$k$ metrics ($k = 1, 5, 10$) and mean Average Precision (mAP) as evaluation metrics. The higher values of the above three metrics, the better performance.

**Implementation Details** OLTM uses a pre-trained CLIP-ViT-B/16 [40] as image encoder, and extracts text features with a pre-trained CLIP Text Transformer. Fine-grained text attributes are derived from a ViLT-based [60] VQA model. Importantly, our VQA model is replaceable. To avoid the cost overhead from multiple calls to VQA model during training and inference, text attribute generation is performed during the data processing stage. For multi-query scenarios, we input a weighted sum of multiple sketches. Input images are resized to $288 \times 144$, and augmented with random horizontal flipping and style augmentation [61]. More experimental configuration details are available in the supplementary materials.

## 5.2 Comparison with State-of-the-Art Methods

**Performance on Market-Sketch-1K.** The experimental results for Market-Sketch-1K are shown in Tab. 1. OLTM significantly outperforms all state-of-the-art methods in both single-query and multi-query settings. In the single-query scenario, OLTM achieves an mAP of 38.35% and a Rank-1 of 36.75%, surpassing state-of-the-art method by 3.38% and 5.23%, respectively. In the multi-query scenario, OLTM achieves an mAP of 62.55% and a Rank-1 of 69.48%, exceeding state-of-the-art methods by 7.37% and 12.85%, respectively. This result is primarily due to the introduction of textual semantic information in the training process of OLTM, which implicitly guides images and sketches to focus on modality-invariant features. Through hierarchical and multi-granularity alignment, the model is able to uncover discriminative fine-grained information, leading to more

Table 3: Ablation studies on Market-Sketch-1K dataset. Training and testing are under the multi-query setting. "Handcrafted" and "VQA" denote manually annotated and VQA generated text attributes, respectively. "Template" represents the sentence template defined by experts. "Prompt" denotes the learnable text prompts. The 'Baseline' uses an image encoder to process both modalities and employs simple cross-attention to integrate the global features. '$\mathcal{L}_{htl}$' [67] represents the hard triplet loss. **Bold** values represent the optimal results.

| Prompt setting | | | | Module | | | Loss | | | Metrics | |
| --- | --- | --- | --- | --- | --- | --- | --- | --- | --- | --- | --- |
| Handcrafted | VQA | Template | Prompt | Baseline | TCA | CFI | $\mathcal{L}_{ID}$ | $\mathcal{L}_{htl}$ | $\mathcal{L}_{tal}$ | mAP | Rank@1 |
| - | - | - | - | | | | | | | 55.47 | 60.04 |
| ✓ | - | ✓ | - | ✓ | ✓ | ✓ | ✓ | - | ✓ | 61.46 | 68.07 |
| ✓ | - | - | ✓ | | | | | | | 61.81 | 67.47 |
| - | ✓ | ✓ | - | | | | | | | 61.76 | 65.46 |
| - | ✓ | - | ✓ | ✓ | - | - | ✓ | - | ✓ | 57.74 | 60.84 |
| | | | | ✓ | ✓ | - | | | | 61.10 | 65.66 |
| - | ✓ | - | ✓ | ✓ | ✓ | ✓ | ✓ | - | - | 54.93 | 57.83 |
| | | | | ✓ | ✓ | ✓ | ✓ | ✓ | - | 61.63 | 66.06 |
| - | ✓ | - | ✓ | ✓ | ✓ | ✓ | ✓ | - | ✓ | **62.55** | **69.48** |

accurate queries. The visualization results on Market-Sketch-1K are shown in Fig. 3. When sketch possesses clearly distinguishable features, query results are satisfactory. In contrast, when sketch closely resembles multiple images, making its features challenging to distinguish with the naked eye, the model encounters errors that are justifiable. In addition, to test the generalization capability of OLTM on unknown styles, cross-style retrieval evaluation is performed on Market-Sketch-1K, and detailed results are in supplementary material.

**Performance on PKU-Sketch.** Tab. 2 presents the model's performance on PKU-SKetch dataset. The results indicate that our OLTM outperforms all competitors by a significant margin. For example, mAP and Rank-1 of OLTM are remarkably high at 91.4 and 94.0, surpassing state-of-the-art method by 5.2% and 4.0%, respectively. Because the sketches on PKU-Sketch contain more detailed information, they can assist multi-granularity interaction in acquiring more fine-grained knowledge. Fig. 3 illustrates the top-5 visualization results of OLTM on PKU-Sketch. Our method can accurately identify the target pedestrians despite challenges such as variations in posture, viewpoint, occlusion, sketch abstraction, and different painting styles.

## 5.3 Ablation Study

In this section, ablation experiments are conducted on Market-Sketch-1K to evaluate the effectiveness of each component within the OLTM framework.

**The Effectiveness of Text Prompt Reasoning.** To evaluate the contribution of different prompt setting, we train the model with different combination of each setting. As shown in Tab. 3, the combination of VQA and Prompt brings significant contribution. Compared to the strategy of directly aligning different modalities, leveraging TPR to implicitly guide the alignment results in improvements of 7.08% in mAP and 9.44% in rank-1. This significant improvement is primarily attributed to introducing textual information during the training phase, which enables the model to effectively capture semantic correlations between images and sketches during inference. Furthermore, compared to manually annotated fixed attributes, those generated by VQA enhance the model's performance by adaptively adjusting the level of detail on which it focuses. Moreover, the introduction of learnable prompts improved the mAP and rank-1 by 0.79% and 4.02%, respectively, compared to fixed templates. Prompt learning can enhance the network's learning and reasoning capabilities, allowing it to more flexibly adapt to diverse modalities and enhance its sensitivity to subtle distinctions.

**The Effectiveness of Our Designed Modules.** To validate the effectiveness of the TCA and CFI module, we progressively integrate them into the baseline and evaluate performance. The results in Tab. 3 indicate that both modules significantly enhance the alignment and interaction capabilities of model across modalities. Specifically, TCA improves mAP and rank-1 by 3.36% and 4.82% compared to baseline, respectively. The introduction of textual information in TCA effectively provides semantic guidance for coarse-grained alignment between modalities, enabling the model

to focus more on similar semantic relationships. Furthermore, the integration of CFI has increased mAP and rank-1 by 1.45% and 3.82%, respectively. CFI selectively focuses on key regions in visual concept representations through semantic consensus. This process optimizes feature interaction and ensures capturing more detailed information at a fine-grained level.

**The Effectiveness of Triplet Assignment Loss.** The experimental results in Tab. 3 demonstrate that the combination of proposed TAL $\mathcal{L}_{tal}$ and identity loss $\mathcal{L}_{ID}$ achieves optimal performance. The identity loss ensures that the model correctly identifies different individual identities, while TAL further optimizes feature space by pulling positive samples closer and pushing negative samples apart. Additionally, to verify the generalization of TAL, as shown in Tab. 4, we achieve superior performance across various network frameworks by substituting TAL for the weighted regularization triplet loss (WRT) [59] and hard triplet loss (HTL). Balancing between Euclidean distance and optimal transport distance can significantly enhance model performance. Please refer to the supplementary materials for more verification experiments on the key role of overall data distribution in enhancing sample feature distance.

Table 4: Performance of TAL $\mathcal{L}_{tal}$ with various baselines. '+' represents WRT; '*' represents HTL $\mathcal{L}_{htl}$.

| Methods | mAP | R@1 |
|---|---|---|
| BDG$^+$ | 24.45 | 24.70 |
| BDG + TAL | **27.79** | **27.71** |
| baseline$^*$ | 57.74 | 60.84 |
| baseline + TAL | **58.41** | **61.04** |
| OLTM$^*$ | 61.63 | 66.06 |
| OLTM + TAL | **62.55** | **69.48** |

## 6 Conclusion

In this paper, we present a optimal transport-based labor-free text prompt modeling (OLTM) framework for sketch re-identification. OLTM embeds text prompt reasoning module and distance measurement into transformer for achieving hierarchical multi-granularity alignment through the guidance of text semantics, leveraging the advantages of prompt learning and optimal transport. In addition, to address the limitations of Euclidean distance in measuring sample similarity, we propose a triplet assignment loss that guarantees a more effective standard based on the overall data distribution. Extensive experiments conducted on two public datasets indicate outstanding performance compared to other state-of-the-art methods for sketch Re-ID.

**Discussion:** In this work, we employ text injection and sample distance optimization to direct the model's attention toward key details, thereby minimizing performance losses due to modal gap and sample abstraction. However, our experiments revealed that when confronted with extremely vague sketch samples (i.e., those that human cannot discern features or match), the model's identification process deteriorates into a random selection among multiple potential outcomes. Therefore, sketch Re-ID heavily depend on the quality of sketches. Enhancing the discriminability of sketches without incurring additional labor costs is a topic worthy of further exploration in future research.

## Acknowledgments

This work was supported in part by the National Natural Science Foundation of China (62272133), in part by the Shenzhen Science and Technology Program (KJZD20230923114600002), in part by the Shenzhen Colleges and Universities Stable Support Program (GXWD20220811170100001), in part by the Key Laboratory of Industrial Equipment Quality Big Data (2024-IEQBD-01), in part by Guangdong Basic and Applied Basic Research Foundation (2024A1515030213).

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

# A  Details about Optimal Transport

This section mainly introduces optimal transport and its corresponding algorithm, Sinkhorn-Knopp [44]. Let $\Gamma_r := \{x \in \mathbb{R}_+^r | x^\top 1_r = 1\}$ represents the probability simplex, where $1_r$ is the $r$-dimensional vector of ones. Given two probability simplex vectors $\alpha \in \Gamma_m$ and $\beta \in \Gamma_n$ and a cost matrix $C \in \mathbb{R}^{m \times n}$, the objective of OT is to seek the optimal transport plan $P^*$ mapping $\alpha$ to $\beta$ at the minimum cost:

$$d_C(\alpha, \beta) = \min_{P \in U(\alpha, \beta)} \langle C, P \rangle,$$
$$U(\alpha, \beta) = \left\{ P \in \mathbb{R}_+^{m \times n} \mid P 1_n = \alpha, P^\top 1_m = \beta \right\}, \tag{7}$$

where $U(\alpha, \beta)$ denotes the transport polytope of $\alpha$ and $\beta$, i.e., the solution space of $P$. The above problem is to find optimal solution $P^*$ in a set of all possible joint probabilities of $(X, Y)$, where $X$ and $Y$ represent random variables with marginal distribution $\alpha$ and $\beta$.

Generally, optimal transport (OT) problem is a linear programming problem that can theoretically be solved in polynomial time. However, in the actual solving process, it involves the square of anchor feature dimensions at all scales, requiring near-cubic complexity [68]. Therefore, consider optimizing the problem through iteration. This method optimizes the solving process by adding an entropy constraint to the OT problem, transforming Eq. 7 into a non-linear convex form with a regularization term:

$$d_C(\alpha, \beta) = \min_{P \in U(\alpha, \beta)} \langle C, P \rangle + \delta E(P),$$
$$U(\alpha, \beta) = \left\{ P \in \mathbb{R}_+^{m \times n} \mid P 1_n = \alpha, P^\top 1_m = \beta \right\}, \tag{8}$$

where $\delta$ is a hyper-parameter and $E(P) = P(\log P - 1)$ is the entropy of $P$. Introducing a regularization term is equivalent to introducing a prior knowledge: without considering the cost matrix $C$, the distribution of assignment matrix $P$ is expected to be as uniform as possible. Eq. 8 is an entropy-regularized OT (EOT) problem and can be solved by Sinkhorn-Knopp algorithm [44] based on iterative updates of vectors. According to the Lagrange Multiplier Method, the conditional extremum problem (i.e., Eq. 8) can be transformed into an unconditional extremum problem:

$$\hat{d}_C(\alpha, \beta) = \min_P \langle C, P \rangle + \delta E(P) + \mu(P 1_n - \alpha) + \rho(P^\top 1_m - \beta) \tag{9}$$

where, $\mu$ and $\rho$ are the Lagrange multipliers. If we take its derivative and set it to 0, we can find $P^*$:

$$P^* = exp(-\frac{\mu}{\delta}) exp(-\frac{C}{\delta}) exp(-\frac{\rho}{\delta}) \tag{10}$$

Let $u = exp(-\frac{\mu}{\delta})$, $v = exp(-\frac{\rho}{\delta})$ and $K = exp(-\frac{C}{\delta})$, two constraint of Eq. 8 needs to be met simultaneously. Thus, one possible solution is to iterate enough times according to the following iteration formula:

$$u^z = \frac{\alpha}{K v^{z-1}}, v^z = \frac{\beta}{K^\top u^z}. \tag{11}$$

Eq. 11 is called as Sinkhorn-Knopp iteration. After $z$ iterations, $P^*$ can be obtained by the following equation:

$$P^* = Diag(u) K Diag(v) \tag{12}$$

It is worth noting that the process of updating $u$ and $v$ (i.e., Eq. 11) alternately can be simplified to a single step: $u^z \leftarrow \alpha / K(\beta / K^\top u^{z-1})$. Furthermore, when all elements of $\alpha$ are positive, this single step can be further simplified as: $u^z \leftarrow 1./\hat{K}(\beta / K^\top u^{z-1})$, where $\hat{K} = Diag(1./\alpha) K$. The overall flow of Sinkhorn-Knopp is described by Algorithm 1.

# B  Details about Triplet Assignment Loss

## B.1  Derivation for Gradient

This appendix provides some details on gradient derivation. To simplify representation and analysis, we focus on a single direction, following the approach in [69], considering that sketch-to-image retrieval and image-to-sketch retrieval are symmetrical. Moreover, we assume that there is only one

---

**Algorithm 1** The OT problem via Sinkhorn-Knopp

---

**Require:** Cost matrix $C$, weights $\alpha$, $\beta$
**Require:** Hyper-parameter $\delta$, iterations $i = 1$, max iteration $z$
 1: Initialize $K = e^{-C/\delta}$, $\hat{K} = Diag(1./\alpha)K$, $u_0 = 1_n/n$
 2: **while** $u$ changes or $i$ is less than or equal $z$ **do**
 3:    $u^i \leftarrow 1./\hat{K}(\beta/K^\top u^{i-1})$
 4:    $i \leftarrow i + 1$
 5: **end while**
 6: Get optimal value $u^* = u$
 7: $v^* = \beta/K^\top u^*$
 8: **return** optimal flow matrix $P^* = Diag(u^*)KDiag(v^*)$

---

paired image for each sketch in the mini-batch. Consequently, we can simplify TRL and TAL as shown below:

$$\mathcal{L}_{trl}(R_i, S_i) = [m - r_i^\top s_i + \hat{r}_i^\top s_i]_+,$$

$$\mathcal{L}_{tal}(R_i, S_i) = [m - d(r_i, s_i) + d(\hat{r}_i, s_i))]_+, \ d(r, s) = \gamma r^\top s + (1 - \gamma)(1 - exp(-\tfrac{r^\top s}{\delta}))r^\top s \tag{13}$$

where $\delta$ is the hyper-parameter of OT problem, $\hat{r}_i$ and $r_i$ are the hardest negative sample and positive sample of the anchor sample $s_i$, respectively. These $l_2$-normalized features are embedded by the modality-specific models, i.e., $f_{\theta_r}(\cdot)$ and $f_{\theta_s}(\cdot)$. Due to the truncation operation $[x]_+$, we only discuss the case of $\mathcal{L} > 0$ that could generate gradients. For TRL, the gradients to the parameters $\theta_r$ and $\theta_s$ are:

$$\frac{\partial \mathcal{L}_{trl}}{\partial \theta_s} = \frac{\partial \mathcal{L}_{trl}}{\partial s_i}\frac{\partial s_i}{\partial \theta_s}, \quad \frac{\partial \mathcal{L}_{trl}}{\partial \theta_r} = \frac{\partial \mathcal{L}_{trl}}{\partial r_i}\frac{\partial r_i}{\partial \theta_r} + \frac{\partial \mathcal{L}_{trl}}{\partial \hat{r}_i}\frac{\partial \hat{r}_i}{\partial \theta_r}. \tag{14}$$

Since the learning of normalized features can be viewed as the movement process of points on a unit hyperplane, we only consider the loss gradients with respect to $r_i$, $\hat{r}_i$ and $s_i$ are:

$$\frac{\partial \mathcal{L}_{trl}}{\partial r_i} = -s_i, \quad \frac{\partial \mathcal{L}_{trl}}{\partial s_i} = \hat{r}_i - r_i, \quad \frac{\partial \mathcal{L}_{trl}}{\partial \hat{r}_i} = s_i. \tag{15}$$

For our TAL, the gradients to the parameters $\theta_r$ and $\theta_s$ are:

$$\frac{\partial \mathcal{L}_{tal}}{\partial \theta_s} = \frac{\partial \mathcal{L}_{tal}}{\partial s_i}\frac{\partial s_i}{\partial \theta_s}, \quad \frac{\partial \mathcal{L}_{tal}}{\partial \theta_r} = \frac{\partial \mathcal{L}_{tal}}{\partial r_i}\frac{\partial r_i}{\partial \theta_r} + \frac{\partial \mathcal{L}_{tal}}{\partial \hat{r}_i}\frac{\partial \hat{r}_i}{\partial \theta_r}. \tag{16}$$

Since the result of $exp(-\tfrac{r^\top s}{\delta})$ is obtained through several iterations of Sinkhorn-Knopp algorithm, this part does not conduct gradients and can be simplified as a coefficient $\hat{\delta} = 1 - exp(-\tfrac{r^\top s}{\delta}) \in [0, 1]$. Thus, the gradients for $r_i$, $\hat{r}_i$ and $s_i$ are:

$$\frac{\partial \mathcal{L}_{tal}}{\partial r_i} = -[\gamma + (1 - \gamma)\hat{\delta}]s_i, \quad \frac{\partial \mathcal{L}_{tal}}{\partial \hat{r}_i} = [\gamma + (1 - \gamma)\hat{\delta}]s_i,$$

$$\frac{\partial \mathcal{L}_{tal}}{\partial s_i} = \gamma(\hat{r}_i - r_i) + (1 - \gamma)\hat{\delta}(\hat{r}_i - r_i). \tag{17}$$

### B.2 The Effectiveness of Triplet Assignment Loss

We have included visual analysis in Fig. 4, which illustrates the convergence curve and sample distances during training. Figure 1(a) shows that conventional triplet loss converges prematurely. In contrast, our proposed triplet assignment loss exhibits higher volatility, reducing the risk of suboptimal local minima. Additionally, Figure 1(b) shows that a specific sketch sample (red box in the top left image) may have similar Euclidean distances to multiple RGB samples. The triplet assignment loss comprehensively considers the distribution of all samples (red box in the lower right image), offering broader possibilities for selecting the most relevant ones.

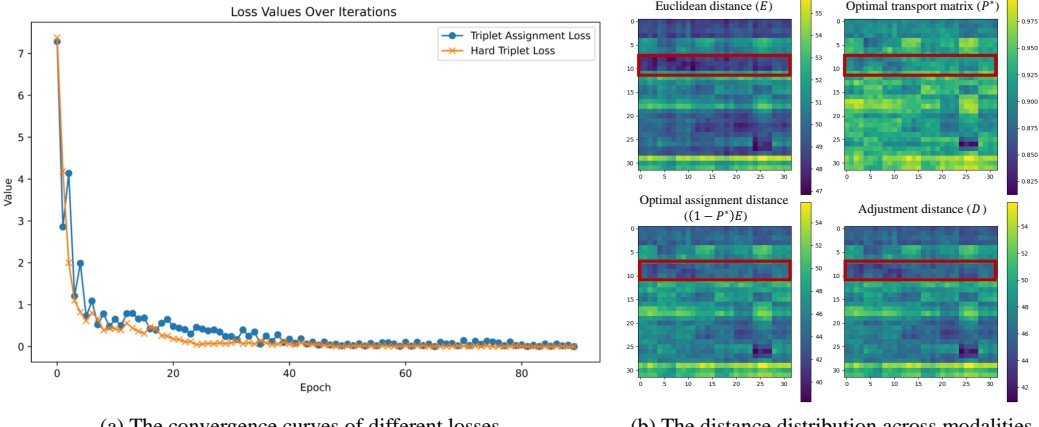

(a) The convergence curves of different losses        (b) The distance distribution across modalities

Figure 4: The effectiveness analysis of Triplet Assignment Loss. In Figure (b), the vertical axis represents RGB images, and the horizontal axis represents sketches. For each ID, 4 training examples are sampled, so the 4x4 cells on the diagonal represent positive sample pairs.

## C  Computational Complexity

Our OLTM achieves the trade-off between performance enhancement and computational complexity. To this end, we select several methods for comparing parameters, floating-point operations (FLOPs), and frames per second (FPS), as shown in the Tab. 5 below. The results show that OLTM gets remarkable performance while maintaining reasonable computational costs. The reason is that: 1) only TCA module is required for inference; 2) the Visual Question Answering(VQA) model is used during data processing.

Table 5: The number of parameters, FLOPs, and FPS of different methods, where **bold** indicates the best performance in this field and underline indicates the second-best performance. VI denotes visible-infrared person re-identification.

| Methods | Field | Backbone | Reference | Paras(M) | FLOPs(G) | FPS |
|---------|-------|----------|-----------|----------|----------|-----|
| DDAG[53] | VI | ResNet50 | ECCV'2020 | 95.6 | **5.2** | 13.2 |
| CM-NAS[54] | VI | ResNet50 | ICCV'2021 | **24.5** | **5.2** | 14.8 |
| CAJ[55] | VI | ResNet50 | ICCV'2021 | 71.6 | **5.2** | 13.1 |
| DEEN[56] | VI | ResNet50 | CVPR'2023 | 89.3 | 13.8 | 7.7 |
| CCSC[9] | Sketch | ViT | MM'2022 | 203.9 | 383.8 | - |
| BDG[6] | Sketch | ResNet50 | MM'2023 | 222.9 | **5.2** | **14.9** |
| UNIReID[7] | Unified | CLIP | CVPR'2023 | **149.6** | 9.3 | 8.2 |
| OLTM (Ours) | Sketch | CLIP | - | 181.9 | 6.2 | 11.7 |

## D  Additional Experimental Results

### D.1  Experimental Setup

**Implementation Details**  The dimensions of image and text features are set to 512. Within a batch, we randomly select 8 identities, each comprising 4 images and 4 sketches. Each image is associated with 9 fine-grained textual attributes. To ensure more reliable comparisons, the random seeds are all set to 0. In the Text Prompt Reasoning module, ConMLP consists of a stack of $N = 2$ identical MLPs, where $\theta_{mlp}$ represents the trainable parameters. The iteration number of the Optimal Transport algorithm is 3. In the Triplet Assignment Loss, the iteration number of the Optimal Transport algorithm is 50. The model is trained with the Adam optimizer, starting with a learning rate of 1e-5, decaying with a cosine scheduler. The model is implemented in PyTorch on the RTX 4090 24GB GPU.

**Cross-style Retrieval**   Given the extensive Market-Sketch-1K dataset, where each person is sketched by six different artists, notable variations exist across these sketches. Consequently, we devise this experiment to assess our model's resilience to diverse artistic styles. Experimental setups involve sketches labeled as S1 to S6, each originating from different artists. Models are trained on sketches by specific artists and tested on sketches by others. And, "single query" denotes separate queries for sketches by different artists of the same individual, while "multi query" indicates queries combining multiple sketches of the same person. The specific details of these experiments are presented in Tab. 6.

Table 6: Testing on unseen styles. We report the mAP score.

a) single-query train and single-query test

| train \ test | $S_6$ | $S_{5,6}$ | $S_{4...6}$ | $S_{3...6}$ | $S_{2...6}$ |
|---|---|---|---|---|---|
| $S_1$ | 25.83 | 22.41 | 22.71 | 23.47 | 22.96 |
| $S_{1,2}$ | 27.10 | 22.72 | 23.18 | 26.47 | – |
| $S_{1...3}$ | 25.75 | 26.60 | 26.83 | – | – |
| $S_{1...4}$ | 32.60 | 30.07 | – | – | – |
| $S_{1...5}$ | 33.87 | – | – | – | – |

b) multi-query train and multi-query test

| train \ test | $S_6$ | $S_{5,6}$ | $S_{4...6}$ | $S_{3...6}$ | $S_{2...6}$ |
|---|---|---|---|---|---|
| $S_1$ | – | – | – | – | – |
| $S_{1,2}$ | – | 31.82 | 33.68 | 39.49 | – |
| $S_{1...3}$ | – | 34.30 | 36.49 | – | – |
| $S_{1...4}$ | – | 39.42 | – | – | – |
| $S_{1...5}$ | – | – | – | – | – |

c) single-query train and multi-query test

| train \ test | $S_6$ | $S_{5,6}$ | $S_{4...6}$ | $S_{3...6}$ | $S_{2...6}$ |
|---|---|---|---|---|---|
| $S_1$ | – | 29.17 | 30.35 | 34.65 | 34.49 |
| $S_{1,2}$ | – | 31.23 | 34.34 | 41.05 | – |
| $S_{1...3}$ | – | 34.52 | 37.65 | – | – |
| $S_{1...4}$ | – | 40.10 | – | – | – |

d) multi-query train and single-query test

| train \ test | $S_6$ | $S_{5,6}$ | $S_{4...6}$ | $S_{3...6}$ | $S_{2...6}$ |
|---|---|---|---|---|---|
| $S_{1,2}$ | 27.10 | 22.68 | 23.35 | 24.81 | – |
| $S_{1...3}$ | 26.97 | 25.13 | 31.96 | – | – |
| $S_{1...4}$ | 29.99 | 27.79 | – | – | – |
| $S_{1...5}$ | 31.96 | – | – | – | – |

**Multi-query Setting**   Similar to [6], "multi-query" involves combining multiple sketches of the same ID during both training and inference. Our paper employs a straightforward fusion method by averaging the image features from multiple sketches. Tab. 7 below provides a comparative analysis of various fusion strategies. The results demonstrate that the basic and simple fusion method achieves the best experimental performance.

Table 7: Performance comparison of different multi-query experimental methods.

| Methods | mAP | Rank@1 | Rank@5 | Rank@10 |
|---|---|---|---|---|
| Simple Fusion | **62.55** | **69.48** | **90.36** | **95.18** |
| Average Pooling | 60.95 | 66.27 | 88.15 | 94.38 |
| Non-local Attention | 60.98 | 65.66 | 90.16 | 94.98 |

**Overfitting Analysis**   To mitigate overfitting, we apply various data augmentation techniques, including random cropping, rotation, and style augmentation. Furthermore, to validate the model's robustness and generalization, we conduct supplementary evaluation experiments on two large-scale datasets (SYSU-MM01 and RegDB) for visible-infrared person re-ID, as shown in Tab. 8. The results demonstrate that our OLTM achieves comparable performance in the visible-infrared domain.

**Parameter Analysis**   For the Triplet Assignment Loss proposed in our work, we compute the sample distance using Eq. 18. Fig. 5 illustrates an analysis of the hyper-parameter $\gamma$. It can be observed that setting $\gamma = 0.3$ yields the best performance in Rank-1 and mAP.

$$D(R_i, S_i) = \gamma E(R_i, S_i) + (1 - \gamma)(1 - P_{i,i}^*)E(R_i, S_i) \tag{18}$$

where $E(R_i, S_i) = \|R_i' - S_i'\|_2$ denotes the Euclidean distance between feature representations.

In Dynamic Consensus Acquisition, the cost matrix $C \in \mathbb{R}_+^{n \times m}$ can be calculated for assignment, where the $(i, j)$-th element $C_{i,j}$ represents the cost of assigning a feature to an atom. Concretely, to introduce certain priors as guidance [23], the enhanced features are used to learn the cost matrix through two fully connected layers initialized randomly. $m$ is a hyper-parameter that needs to be set. Fig. 6 analyzes the values of $m$, showing that the best performance in terms of Rank-1 and mAP is achieved when $m = 32$.

Table 8: Comparison results of our method on visible-infrared datasets, namely SYSU-MM01 and RegDB. Market-Sketch-1K is a sketch dataset used for reference. "VI" and "Sketch" represent their respective task domains.

| Methods | Domain | SYSU-MM01 | | | | RegDB | | | | Market-Sketch-1K | |
| | | All Search | | Indoor Search | | VIS to IR | | IR to VIS | | Sketch to VIS | |
| | | R-1 | mAP | R-1 | mAP | R-1 | mAP | R-1 | mAP | R-1 | mAP |
|---|---|---|---|---|---|---|---|---|---|---|---|
| DDAG[53] | VI | 54.8 | 53.0 | 61.0 | 68.0 | 69.3 | 63.5 | 68.1 | 61.8 | 11.2 | 12.1 |
| CM-NAS[54] | VI | 60.8 | 58.9 | 68.0 | 52.4 | 82.8 | 79.3 | 81.7 | 77.6 | 0.7 | 0.8 |
| CAJ[55] | VI | 69.9 | 66.9 | 76.3 | 80.4 | 85.0 | 79.1 | 84.8 | 77.8 | 1.5 | 2.4 |
| DART[57] | VI | 68.7 | 66.3 | 72.5 | 78.2 | 83.6 | 75.7 | 82.0 | 73.8 | 6.6 | 7.8 |
| DCLNet[58] | VI | 70.8 | 65.3 | 73.5 | 76.8 | 81.2 | 74.3 | 78.0 | 70.6 | 12.2 | 13.5 |
| OLTM(Ours) | Sketch | 70.6 | 68.6 | 76.2 | 80.4 | 84.8 | 77.2 | 83.9 | 75.5 | 36.8 | 38.4 |

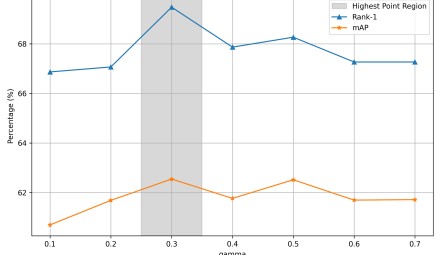

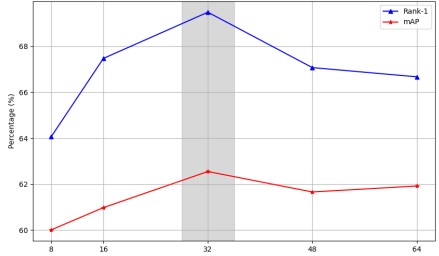

Figure 5: Analysis of the hyperparameter $\gamma$     Figure 6: Analysis of the hyperparameter $m$

**Qualitative Analysis**   Fig. 7 shows the Rank-5 results of OLTM and the baseline on the Market-Sketch-1K dataset. The left and right parts show the retrieval results of the baseline and OLTM, respectively. We can observe that OLTM can focus on more fine-grained discriminative information, such as bag and hat. In contrast, the baseline only considers global information matching, which leads to performance degradation.

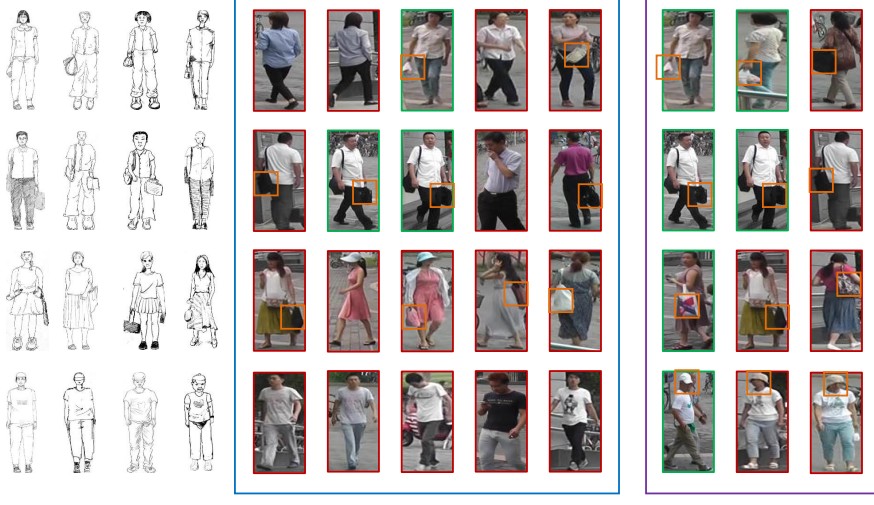

Figure 7: The Rank-5 retrieval results under the multi-query setting on the Market-Sketch-1K dataset are presented. On the left side are the retrieval results of the baseline, and on the right side are the retrieval results of OLTM. Green borders indicate successful retrieval of the target pedestrian, while red borders indicate incorrect results. Yellow boxes represent fine-grained information.

# E  Analysis of VQA Models

## E.1  The Setup of the VQA Problems

Through extensive statistical analysis, we have formulated 9 specific questions to obtain the corresponding fine-grained textual attributes for a given image. These nine questions are as follows:

1. What is the gender of this person?
2. Is this person with long or short hair?
3. What is the color of this person's shirt?
4. Is this person wearing long sleeves or short sleeves?
5. Is this person wearing pants or a dress underneath?
6. What is the color of this person's lower garment?
7. Is this person carrying a backpack?
8. Is this person wearing a hat?
9. Is this person wearing glasses?

## E.2  Replaceability of VQA Model

The VQA model is inherently substitutable. Essentially, any visual-language model which is capable of generating target attribute information from images can serve as an alternative. We also use other VQA models to demonstrate their substitutability, as shown in the Tab. 9 below.

Table 9: Performance comparison of different VQA models.

| Methods | mAP | R@1 | R@5 | R@10 |
|---|---|---|---|---|
| BLIP [70] | 62.63 | 67.87 | 91.37 | 96.79 |
| GIT [71] | 62.33 | 68.47 | 90.76 | 96.59 |
| VILT (ours) | 62.55 | 69.48 | 90.36 | 95.18 |

## E.3  The Fine-grained Recognition Ability of VQA Model

The VQA model has the capability to describe fine-grained recognition information for the following reasons: 1) The VQA model generates detailed attributes about various aspects of the pedestrian target (e.g., hair, backpack, hat), rather than relying on complete descriptive sentences. 2) We provide a visualization comparison, as shown in Fig. 8. This comparison demonstrates that using text attributes can guide effectively the attention of model.

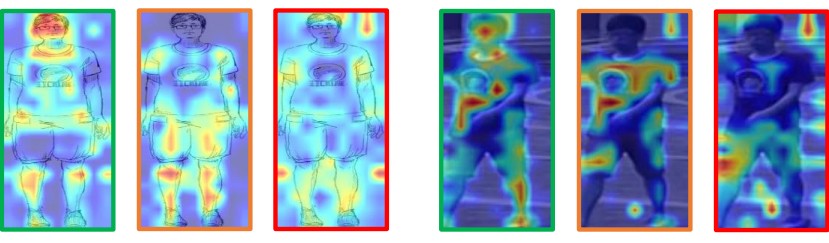

Sketch Modality          Visible Modality

Figure 8: Visualization of attention maps. The **Green** and **Red** box indicate presence and absence of text attribute guidance, respectively. The **Orange** box represents partial text attribute guidance (excluding head-related information).

## E.4 Background Information

We conduct additional background evaluation on Market-Sketch-1k dataset based on our initial studies. Specifically, we formulate the question *'What is the background of this image?'* to extract textual attributes about background. The extracted background details are illustrated in Fig. 9. However, the introduction of background information result in a decrease of 2.81 in Rank-1 and 1.62 in mAP. This decline can be attributed to the absence of corresponding background information in the sketches, which potentially interferes with the model's learning process.

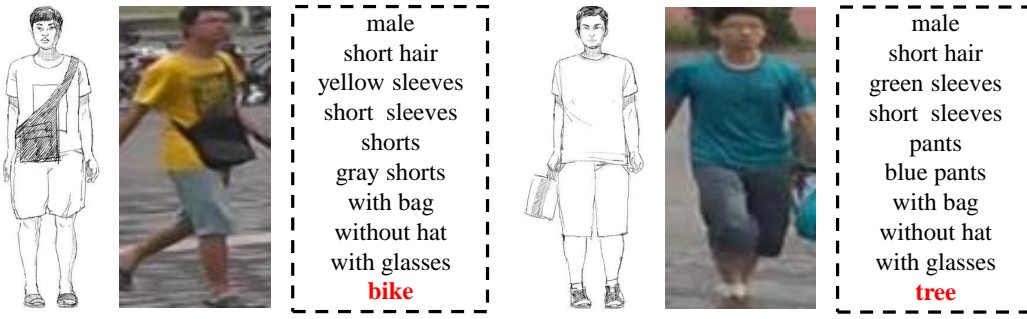

Figure 9: The text attributes generated by VQA model on RGB images. **Red** indicates background information obtained from the question: "What is the background of this image?".

## E.5 The Availability of Text Attributes

To verify the effectiveness of different text attributes, we have provided additional ablation experiments in Tab. 10 below. The results show a significant decrease in model performance after discarding several hard-distinguished attributes (e.g., color and gender) in sketches. As shown in Fig. 10, sketches convey gender-related information through factors like body shape, and the contrast between light and dark areas effectively highlights specific color details. The TPR module injects detailed information into modal interactions during training. This enables the model to focus on these nuances autonomously, even without TPR during inference.

Table 10: Building on the original experimental setup, this comparison evaluates performance by removing the fine-grained textual attributes of gender, up color, and down color.

| gender | up color | down color | mAP | Rank@1 | Rank@5 | Rank@10 |
|:---:|:---:|:---:|:---:|:---:|:---:|:---:|
| ✓ | ✓ | ✓ | 62.55 | 69.48 | 90.36 | 95.18 |
| - | ✓ | ✓ | 62.28 | 68.07 | 89.96 | 95.18 |
| - | - | ✓ | 61.64 | 67.67 | 89.76 | 94.98 |
| - | - | - | 61.35 | 67.07 | 89.16 | 94.78 |

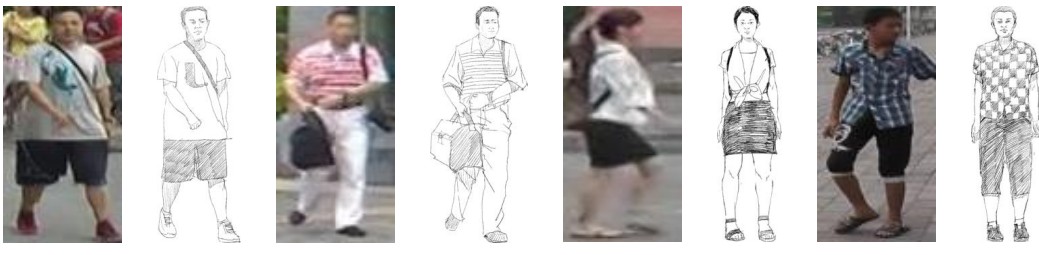

Figure 10: Comparative analysis of RGB images and corresponding sketches. Gender-related factors in sketches include body shape and hair, and the contrast highlights color details.

