# OpenReview forum: "Optimal Transport-based Labor-free Text Prompt Modeling for Sketch Re-identification"
_NeurIPS.cc/2024/Conference — NeurIPS 2024 poster_

### Official Review · Reviewer_beT2 · 2024-07-06

**Soundness:** 3
**Presentation:** 3
**Contribution:** 3
**Rating:** 6
**Confidence:** 5

**Summary:**

This article extends image-based person and vehicle reid task to sketch field. First, authors introduce a novel sketch-based reid framework named OLTM, which utilizes text information to achieve modal alignment. Additionally, in sketch-based reid, authors apply the VQA model to generate textual attributes of persons, thus avoiding costly annotation efforts. Furthermore, authors introduce a novel triplet alignment loss to enhance distance calculation.

**Strengths:**

1. The authors use the VQA model to generate attribute descriptions about pedestrians, effectively reducing the cost of manual labeling.

2. The method utilizes optimal transport theory and text prompt to improve model performance and designs a new triplet alignment loss.

3. The experimental results are convincing compared to the state-of-the-arts.

**Weaknesses:**

1. Compared to CNN-based methods, using CLIP as a Backbone may suffer from high computation complexity.

2. Some minor issues: the symbols for K, Q, and V in Eq. 4 and 5 should be bolded.

3. The names in the paper and the supplementary material are inconsistent, i.e., HTL/TRL.

4. Figure 2 in the supplementary material is rather blurry.

**Questions:**

1. Why is the ViLT-based VQA model used instead of others, like Bert-based model?

2. Is there an error in Eq. 7? After checking the supplementary materials, my understanding is that the optimal transport matrix P* is a weight matrix, not a loss value.

3. Some details are not clear to me. It is unclear whether the process of obtaining text attributes occurs during model training or after pre-processing. According to the framework Figure 2, this process seems to be executed during the model training process, which can result in complex model calls.

4. The experiment doesn’t mention whether it is compared with existing image-based re-identification methods.

---

> ### Author Rebuttal · Authors · 2024-08-07
>
> **W1: Computational complexity.**
>
> **A:** Please refer to **[Respones to nGMK:W1, edWJ:W6, beT2:W1]** in **Common Response**.
>
> **W2, W3, W4: Minor issues.**
>
> **A:** Thank you for your thorough review and for highlighting these minor issues. We have revised the manuscript to address the mentioned issues.
>
> **Q1: Replaceability of VQA model.**
>
> **A:** Please refer to **[Response to edWJ:W7, beT2:Q1]** in **Common Response**.
>
> **Q2: $\boldsymbol{P}^\ast$.**
>
> **A:** Thank you for bringing up this question. Indeed, $P^\ast$ represents a weight matrix. Here is the corrected formula:
>
> \begin{array}{c}
>     \mathcal{L}\_{tal}(R\_i, S\_i) = [m - D(R\_i, S\_i) + D(R\_i, \hat{S}\_h)]\_{+} + [m - D(R\_i, S\_i) + D(\hat{R}\_h, S\_i)]\_{+}, \\\\
>     D(R\_i, S\_i) = \gamma E(R\_i, S\_i) + (1 - \gamma)(1 - \boldsymbol{P}^{\ast}\_{i,i})E(R\_i, S\_i)
> \end{array}
>
> *where, $\boldsymbol{P}^{\ast}$ is the optimal transport matrix with the least amount of cost, and $\boldsymbol{P}^{\ast}\_{i,i}$ represents the assignment weight of $(R\_i, S\_i)$ obtained after balancing the overall distribution.*
>
> **Q3: Details of text attribute acquisition.**
>
> **A:** Thank you for raising this question. The text attributes acquisition occurs **during the data processing stage**, and does not lead to complex model calls during training. We have enhanced clarity in our manuscript to prevent such ambiguities.
>
> **Q4: Experimental comparisons.**
>
> **A:** Thank you for your constructive feedback. Based on task relevance, our adopted methods are all image-based re-identification methods, categorized into visible-infrared and sketch-based methods. The former include DDAG [54], CM-NAS [55], CAJ [56], MMN [57], DART [58], DCLNet [59], DSCNet [60], and DEEN [61]. The latter comprises BDG [6] and UNIReID [7]. We have already given the detailed descriptions of these experimental methods in the manuscript.

---

> > ### Comment · Reviewer_beT2 · 2024-08-14
> >
> > Thanks to the authors for the reply. The authors answered most of my questions. Therefore, I maintain my positive rating.

---

> > > ### Author Response · Authors · 2024-08-14
> > >
> > > Thank you for your time and efforts. We are encouraged that your concerns have been addressed, and we greatly appreciate your positive feedback on our work.

---

### Official Review · Reviewer_Bynq · 2024-07-08

**Soundness:** 3
**Presentation:** 3
**Contribution:** 3
**Rating:** 7
**Confidence:** 5

**Summary:**

This paper focuses on sketch based person ReID. Especifically, a labor free text method with OT is proposed, which achieve a large performance improvement on two public databases. Overall, I think the idea of this paper is interesting and the experiments show its superiority.

**Strengths:**

1. The proposed method is interesting. For instance, authors introduce the text without any labor cost to handle the modal-gap removal. Also, the OT is used to get interested local text representation.
2. The performance on two datasets achieves remarkable improvement.

**Weaknesses:**

1. Some grammar errors should be corrected.
2. More detailed analysis on OT and TPR should be provided.
3. From the paper, I don't quite understand the settings of the parameters. In the proposed method, what is the mean of $\alpha$ and $\beta$? What is the relationship between them and cost matric $C$?

**Questions:**

1. In the related works, there is a subsection on optimal transport. What is the relationship of [26]/[27] and OT.
2. On Line 143 “represents” should be modified to “represent”. On page 6, P* should be bold, be consistent to above.
3. On line 185, authors argue that the proposed strategy would not introduce any noise. So, how to ensure it?
4. In eq.(7), it is better to give more description about how to calculate the P*.

**Limitations:**

The author has addressed the limitation.

---

> ### Author Rebuttal · Authors · 2024-08-07
>
> **W1, Q2: Minor issues.**
>
> **A:** Thanks for your nice comment. We have identified all errors and corrected them in the manuscript.
>
> **W2, W3, Q4: Optimal transport.**
>
> **A:** Thanks for your constructive comment. Due to space constraints, a detailed description of optimal transport (OT) and matrix $\boldsymbol{P^\ast}$ can be found in the supplementary materials. Essentially, the OT problem resembles a "supplier-demander" issue: $\boldsymbol{\alpha}$ denotes the quantity of goods each supplier can provide, $\boldsymbol{\beta}$ denotes the quantity of goods each demander requires, and the cost matrix $\boldsymbol{C}$ indicates the expense incurred when each supplier supplies one unit of goods to each demander. The objective is to find the assignment solution that minimizes costs, yielding the optimal transport matrix $\boldsymbol{P^\ast}$. We add the description about how to calculate the $\boldsymbol{P^\ast}$:
>
> *For an input positive pair $(R\_i, S\_i)$ in a mini-batch $x$, the feature representations obtained through model inference are denoted as $f(R\_i)$ and $f(S\_i)$. If the sets of all sample features from different modalities in $x$ are treated as two discrete distributions, their alignment can be considered an optimal transport problem. The cost matrix $\boldsymbol{\hat{C}}$ is derived from pairwise feature similarities: $\boldsymbol{\hat{C}}\_{i,j}=[f(R\_i)^{\top}f(S\_j)]\_+$. We aim to acquire the optimal transport matrix $\boldsymbol{P}^{\ast}$ with the least amount of cost, where $\boldsymbol{P}^{\ast}\_{i,j}$ represents the assignment weight of $(R\_i, S_j)$ obtained after balancing the overall distribution. TAL can be represented based on triplet loss as the weighted sum of the original distance and the optimal assignment distance, dynamically updated at a certain rate $\gamma$:*
>
> \begin{array}{c}
>     \mathcal{L}\_{tal}(R\_i, S\_i) = [m-D(R\_i, S\_i)+D(R\_i, \hat{S}\_h)]\_{+} + [m-D(R\_i, S\_i)+D(\hat{R}\_h, S\_i)]\_{+},    \\\\
>     D(R\_i, S\_i) = \gamma E(R\_i, S\_i) + (1-\gamma)(1-\boldsymbol{P}^{\ast}\_{i,i})E(R\_i, S\_i)
> \end{array}
>
> *where  $\hat{R}\_h=argmax\_{R\_j\neq{R\_i}}D(R\_j,S\_i)$  and  $\hat{S}\_h=argmax\_{S\_j\neq{S\_i}}D(R\_i,S\_j)$ are the most similar negatives in $x$ for $(R\_i, S\_i)$, $[x]\_{+}=\max(x,0)$,  and $E(R\_i, S\_i)=\Vert{f(R\_i)-f(S\_i)}\Vert\_2$ denotes the Euclidean distance between feature representations.*
>
> **Q1: The relationship between Reference [26]/[27].**
>
> **A:** Both papers [26-27] utilize optimal transport to address inter-modal distribution alignment in re-ID tasks. Ling et al. [26] select an optimal transport strategy and assign high weights to pairs with smaller intra-identity variation. Zhang et al. [27] first attempt to find an optimal transport matrix to re-weight the distance of different local parts in re-ID. These two papers inspire us to apply optimal transport theory for exploring more discriminative feature representations and establishing more reasonable standards for measuring sample distances in re-ID.
>
> **Q3: Noise.**
>
> **A:** Thank you for pointing out the ambiguity in this sentence. Compared to fixed handcrafted prompts, our dynamic knowledge learning mechanism does not rely on expert knowledge. This strategy can mitigate **additional noise caused by inaccurate sentence templates**, since our prompt include adaptable learnable components constrained by various loss constraints. We have revised it in the manuscript: *"This integration introduces a dynamic knowledge learning mechanism that reduces noise introduction compared to handcrafted prompts, while enhancing flexible interaction across modalities and boosting the transferability of text embeddings."*

---

> > ### Comment · Reviewer_Bynq · 2024-08-12
> >
> > Thank you for the author's reply. The detailed reply and analysis have dispelled my doubts about OT, TPR, and parameter settings.

---

> > > ### Author Response · Authors · 2024-08-12
> > >
> > > Thanks for your response and updating the rating. Your feedback is deeply valued and helps improve our paper.

---

### Official Review · Reviewer_edWJ · 2024-07-09

**Soundness:** 4
**Presentation:** 3
**Contribution:** 3
**Rating:** 6
**Confidence:** 5

**Summary:**

This paper proposes an optimal transport-based labor-free text prompt network for sketch re-identification. The authors address two primary challenges: the expense of text annotation and cross-modal interaction, leveraging generated text attributes for multi-granularity modal alignment. The experimental results on two datasets achieve significant improvements.

**Strengths:**

This paper solves a practical problem and proposes a framework setting corresponding to the sketch re-identification task. The experimental results have effectively verified the effectiveness of this setting. If all the data and protocols are available, I believe it will be valuable for the research community.

**Weaknesses:**

The paper is interesting and the topic authors discussed is very promising, but I feel there are some doubts:

1. The author suggests that the classic triplet loss may lead to inaccurate estimates of sample distances and potentially result in suboptimal local minima. The proposed Triplet Assignment Loss aims to address this issue. How can this viewpoint be substantiated from experiments?

2. Comparisons with the most related method [1] are missing in TABLE 1 and TABLE 2.

3. Do the transformer blocks in feature enhancement module for different modalities share weights?

4. The description of pedestrians comes from VQA's answers to 9 questions. These questions typically cover details like color and gender. However, sketches used in testing often lack such explicit information. How does the model then prioritize these aspects?

5. The experimental results in Table 1 include "multiple-query". However, the paper does not provide a description of how this is achieved.

6. The paper compares the structural and performance differences between OLTM and other methods. However, it does not discuss whether the additional structural design increases the computational burden. Including relevant cost comparison experiments would greatly enhance the study.

7. Is the VQA model irreplaceable? I did not come across a statement like "VQA model is optional" in the paper.

[1] Li H, Ye M, Zhang M, et al. All in One Framework for Multimodal Re-identification in the Wild[C]//Proceedings of the IEEE/CVF Conference on Computer Vision and Pattern Recognition. 2024: 17459-17469.

**Questions:**

Major concerns: see Weakness.
Minor concerns:
a)	In 4.1, the inference stage uses Rcls and Scls instead of r and s; the bolding of the formula needs to be checked.
b)	Add detailed descriptions of the cross-style retrieval experiment in the supplementary materials.

**Limitations:**

The author mentions that their proposed framework effectively tackles challenges like occlusion and perspective but still encounters misjudgments with highly similar but distinct individuals. I consider this a critical issue for collective resolution in advancing re-identification tasks. My positive evaluation of the research remains unchanged. I will, however, follow the authors' work and look forward to potential solutions for these challenges.

---

> ### Author Rebuttal · Authors · 2024-08-07
>
> **W1: Triplet Assignment Loss.**
>
> **A1:** Thank you for your inquiry. We have included visual analysis in **Figure 1 of PDF**, which illustrates the convergence curve and sample distances during training. **Figure 1(a)** shows that conventional triplet loss converges prematurely. In contrast, our proposed triplet assignment loss exhibits higher volatility, reducing the risk of suboptimal local minima. Additionally, **Figure 1(b)** shows that a specific sketch sample (red box in the top left image) may have similar Euclidean distances to multiple RGB samples. The triplet assignment loss comprehensively considers the distribution of all samples (red box in the lower right image), offering broader possibilities for selecting the most relevant ones.
>
> **W2: Reference.**
>
> **A2:** We appreciate the reviewer’s recommendation. Inspired by this valuable work, we have cited this reference into Sec. 2 of the manuscript: "... fine-grained interaction. **Furthermore, Li et al. [1] first attempt to conduct in-depth research on zero-shot multimodal ReID through a large foundational model.** In this paper, we ...". However, since the source code of this reference is not available and it mainly focuses on zero-shot learning, we cannot make comparsion with it in experiments.
>
> **W3: Weights sharing.**
>
> **A3:** Thank you for your question. As stated in Sec. 4.3, Transformer blocks for different modalities do not share weights in feature enhancement module. The cross-modal interaction component in front of this module shared weights, facilitating the extraction of common features between modalities. Consequently, this module enables each modality to autonomously refine its representation. This strategy ensures precise adjustments according to specific needs and enhances the overall performance.
>
> **W4: The availability of text attributes.**
>
> **A4:** Thank you for pointing out this comment. To verify the effectiveness of different text attributes, we have provided additional ablation experiments in Table 5 below. The results show a significant **decrease in model performance** after discarding several hard-distinguished attributes (e.g., color and gender) in sketches. As shown in **Figure 3 of PDF**, sketches convey gender-related information through factors like body shape, and the contrast between light and dark areas effectively highlights specific color details. The TPR module injects detailed information into modal interactions during training. This enables the model to focus on these nuances autonomously, even without TPR during inference.
>
> *Table 5: The experiment results on various text attributes. "Gender", "Shirt" and "Pants" indicate whether the attribute includes gender, shirt color and pants color.*
> | Gender | Shirt  | Pants  | mAP | Rank@1 | Rank@5 | Rank@10 |
> | ---- | ---- | ---- | ---- | ---- | ---- | ---- |
> | ✓ | ✓ | ✓ | 62.55 | 69.48 | 90.36 | 95.18 |
> | - | ✓ | ✓ | 62.28 | 68.07 | 89.96 | 95.18 |
> | - | - | ✓ | 61.64 | 67.67 | 89.76 | 94.98 |
> | - | - | - | 61.35 | 67.07 | 89.16 | 94.78 |
>
> **W5: Multi-query.**
>
> **A5:** Thank you for your question regarding the experimental setup. Similar to [2], "multi-query" involves combining multiple sketches of the same ID during both training and inference. Our paper employs a straightforward fusion method by averaging the image features from multiple sketches. Table 6 below provides a comparative analysis of various fusion strategies. The results demonstrate that the basic and simple fusion method achieves the best experimental performance.
>
> *Table 6: Performance comparison of different multi-query experimental methods.*
> | Methods | mAP | Rank@1 | Rank@5 | Rank@10 |
> | ---- | ---- | ---- | ---- | ---- |
> | Simple Fusion | 62.55 | 69.48 | 90.36 | 95.18 |
> | Average Pooling | 60.95 | 66.27 | 88.15 | 94.38 |
> | Non-local Attention | 60.98 | 65.66 | 90.16 | 94.98 |
>
> **W6: Computational complexity.**
>
> **A6:** Please refer to **[Response to nGMK:W1, edWJ:W6, beT2:W1]** in **Common Response**.
>
> **W7: Replaceability of VQA model.**
>
> **A7:** Please refer to **[Response to edWJ:W7, beT2:Q1]** in **Common Response**.
>
> **Q1: Minor concerns.**
>
> **A1:** We apologize for this typing mistake. We have revised it in the manuscript and added the following note to the supplementary material: *"This section provides a detailed explanation of the cross-style retrieval. Given the extensive Market-Sketch-1K dataset, where each person is sketched by six different artists, notable variations exist across these sketches. Consequently, we devise this experiment to assess our model's resilience to diverse artistic styles. Experimental setups involve sketches labeled as S1 to S6, each originating from different artists. Models are trained on sketches by specific artists and tested on sketches by others. And, "single query" denotes separate queries for sketches by different artists of the same individual, while "multi query" indicates queries combining multiple sketches of the same person."*
>
> [1]Li H, Ye M, Zhang M, et al. "All in One Framework for Multimodal Re-identification in the Wild" *CVPR* 2024.
>
> [2]Lin K, Wang Z, Wang Z, et al. "Beyond Domain Gap: Exploiting Subjectivity in Sketch-Based Person Retrieval" *ACMMM* 2023.

---

> > ### Comment · Reviewer_edWJ · 2024-08-09
> >
> > This is a good job, and the author has also answered my questions well.

---

> > > ### Author Response · Authors · 2024-08-12
> > >
> > > Thank you for recognizing our work and raising your score. Your suggestions significantly help improve the quality of our paper.

---

### Official Review · Reviewer_nGMK · 2024-07-09

**Soundness:** 2
**Presentation:** 3
**Contribution:** 2
**Rating:** 4
**Confidence:** 4

**Summary:**

This article proposes a framework for pedestrian re identification in sketch images based on optimal transportation theory, which utilizes visual question answering pre training models to address the current issue of high text annotation costs and only focuses on global feature representation. This framework utilizes a visual question answering model to automatically generate text prompts, and extracts coarse-grained features through interaction between the text encoder and the visual encoder. By combining clustering methods based on optimal transportation theory with multi head attention mechanism, fine-grained features are extracted, achieving hierarchical and multi granularity alignment, and obtaining a relatively complete feature representation. In addition, a triplet loss function was designed that prioritizes regions with maximum local feature similarity, providing a more reasonable representation for measuring local feature distance. Through experiments on two public datasets, it can be proven that the pedestrian re identification and generalization ability of this model framework have been improved compared to previous model methods.

**Strengths:**

(1) Reduced the cost of text annotation. Due to the significant modal differences between sketched images and real images, it is almost inevitable to add an intermediate modality to assist information exchange, while text information is a very practical intermediate modality information. Due to the high cost of manual annotation, the introduction of visual question answering models to supplement intermediate modal information is highly effective.
(2) Layered and multi granularity alignment helps to mine richer sample information. Traditional text prompts mainly focus on global features, while this model uses clustering methods based on optimal theory and multi head attention mechanisms to process fine-grained information after extracting coarse-grained features, and obtains valuable local information from it.
(3) The triplet loss function takes into account the similarity differences between introduced local features. The traditional triplet loss function assigns the same weight to the differences between positive and negative samples, which may lead to inaccurate estimation of the local feature distance of the samples. The model framework proposed in this paper takes into account the distance difference between regions corresponding to fine-grained features, providing a more reasonable measurement method.

**Weaknesses:**

(1) The calculation cost is relatively high. Due to the introduction of a new visual question answering model and the addition of multiple attention mechanism structures, the computational resources required by the model are significantly increased compared to previous models.
(2) Overfitting is prone to occur. During the experiment, it was found that due to the small sample size of the sketch dataset and the rich granularity and depth of the model's feature extraction, overfitting is prone to occur, which can also affect the model's generalization ability.
(3) The accuracy of text prompts is greatly affected by image quality. The text prompts designed in this article have some fixed prompts and some learnable units. Due to the accuracy of text information directly affecting the quality of feature extraction, the performance of visual question answering pre training models on sketch image modalities is easily affected by different styles of images.

**Questions:**

(1) How is the number of learnable text prompt units added to a fixed text prompt vector determined?
(2) Does the visual question answering model have the ability to describe feature information that truly possesses fine-grained recognition ability?
(3) Is there a comparison of models with artificial text prompts in the experimental section? Will this method still have advantages in terms of re identification performance compared to models with manual text prompts that have been described with fine-grained features?
(4) Regarding the use of optimal transportation theory to remove irrelevant information, is it possible to increase recognition accuracy if the text description can involve background information?

**Limitations:**

(1) The performance of visual question answering models on sketch images will limit their ability to extract features, so it is necessary to evaluate whether the VQA model can provide fine-grained local feature descriptions on sketch images;
(2) There may still be misjudgments when performing re identification on two individuals with similar appearances but different identities, so text prompts may need to refer to more information, such as extracting content that is beneficial for judgment from discarded background information.

---

> ### Author Rebuttal · Authors · 2024-08-07
>
> **W1: Computational complexity.**
>
> **A1:** Please refer to **[Response to nGMK:W1, edWJ:W6, beT2:W1]** in **Common Response.**
>
> **W2: Overfitting.**
>
> **A2:** We sincerely appreciate your valuable comments regarding overfitting, as this is also a concern for us. Consistent with the approaches in [1] and [2], we validate our model on two publicly available benchmark datasets, PKU-Sketch and Market-Sketch-1K, to ensure fair comparisons. To mitigate overfitting, we apply various data augmentation techniques, including random cropping, rotation, and style augmentation. Furthermore, to validate the model's robustness and generalization, we conduct supplementary evaluation experiments on two large-scale datasets (SYSU-MM01 and RegDB) for visible-infrared person re-ID, as shown in **Table 1 of PDF**. The results demonstrate that our OLTM achieves comparable performance in the visible-infrared domain.
>
> **W3: Image quality.**
>
> **A3:** Thank you for your insightful comments.
> 1) It should be clarified that the VQA model is only applied to RGB images, not sketch.
> 2) In fact, image quality affects not only the attributes generated by VQA, but also the performance of all visual tasks. Therefore, in addition to several data augmentation operations (e.g., random rotation, flipping, and style augmentation), we have also implemented the following strategies: 1）In contrast to the conventional method of directly generating a complete descriptive sentence，we use a "divide-and-conquer" strategy. By employing multiple questions to capture specific details (e.g., hair and shirt color), we effectively reduce potential noise in the text prompts. 2) Fixed attribute tokens are dynamically embedded into learnable prompts. In addition, applying various loss functions guides the model to prioritize reliable and valuable information. 3) To further refine fine-grained text embeddings, a "bin" mechanism is added to categorize potentially ambiguous content during Dynamic Consensus Acquisition (as analyzed in Sec. 4.2).
>
> **Q1: The number of learnable text prompts.**
>
> **A1**: Thank you for your question. As analyzed in Section 4.2, we extract $k$ text attributes $\\{att_1, att_2,..., att_k \\}$ using CLIP tokenizer to derive text tokens $a_i=Tokenizer(att_i)$, where each $a_i$ contains $n$ fixed numbers of tokens. Subsequently, $a_i$ preserves valid tokens, and is fed into CLIP token encoder to obtain the fixed attribute tokens $a_i \in{\mathcal{R}^{m\times{c}}}$. Here each word can be represented by $m$ valid tokens of dimension $c$ [3]. Those fixed parts are uniformly embedded into $l$ learnable prompts of dimension $c$, and finally we obtain final text description $q$. In our implementation details, the value of $k$ is $9$, and we set $n=77$ based on CLIP's fixed configuration. Thus the number of learnable text prompt is $n-k\times{m}$.
>
> **Q2: The fine-grained recognition ability of VQA model.**
>
> **A2:** Thank you for your valuable comment. The visual question answering (VQA) model has the capability to describe fine-grained recognition information for the following reasons: 1) The VQA model generates detailed attributes about various aspects of the pedestrian target (e.g., hair, backpack, hat), rather than relying on complete descriptive sentences. 2) We provide a visualization comparison, as shown in **Figure 4 of PDF**. This comparison demonstrates that using text attributes can guide effectively the attention of model.
>
> **Q3: The performance comparison with the handcrafted prompts.**
>
> **A3:** Thank you for pointing out the incompleteness in our ablation study. Our method remains competitive compared to handcrafted prompts mechanism[4], as shown in Table 4 below. The performance improvements can be attributed to the following reasons: 1) We introduce a dynamic learnable prompt mechanism without requiring additional expert knowledge. This enhances the model's adaptability and robustness. 2) VQA is more flexible compared to handcrafted prompts. It can generate more detailed information (e.g., glasses, hats) than what is included in handcrafted prompts.
>
> *Table 4: Comparison of prompt setting methods. "Handcrafted" and "VQA" denote manually annotated and VQA generated text attributes, respectively. "Template" represents the sentence template defined by experts. "Prompt" denotes the learnable text prompts.*
> | Handcrafted | VQA | Template | Prompt | mAP   | R@1   | R@5   | R@10  |
> |-|-|-|-|-|-|-|-|
> | ✓ | - | ✓ | - | 61.46 | 68.07 | 89.96 | **96.79** |
> | ✓ | - | - | ✓ | 61.81 | 67.47 | **90.56** | 95.78 |
> | - | ✓ | ✓ | - | 61.76 | 65.46 | 90.16 | 96.18 |
> | - | ✓ | - | ✓ | **62.55** | **69.48** | 90.36 | 95.18 |
>
> **Q4: Background information.**
>
> **A4:** Thanks for your constructive advice. This motivates us to investigate the feasibility of incorporating background information into text descriptions. Accordingly, we conduct additional background evaluation on Market-Sketch-1k dataset based on our initial studies. Specifically, we formulate the question **'What is the background of this image?'** to extract textual attributes about background. The extracted background details are illustrated in **Figure 2 of PDF**. However, the introduction of background information result in **a decrease of 2.81 in Rank-1 and 1.62 in mAP**. This decline can be attributed to the absence of corresponding background information in the sketches, which potentially interferes with the model's learning process.
>
> [1]Pang L, et al. "Cross-domain adversarial feature learning for sketch re-identification" *ACMMM* 2018.
>
> [2]Lin K, et al. "Beyond Domain Gap: Exploiting Subjectivity in Sketch-Based Person Retrieval" *ACMMM* 2023.
>
> [3]Radford A, et al. "Learning transferable visual models from natural language supervision" *ICML* 2021.
>
> [4]Lin Y, et al. "Improving person re-identification by attribute and identity learning. *Pattern recognition* 2019.

---

> > ### Author Response · Authors · 2024-08-14
> >
> > We sincerely want to know if our response has addressed your concerns. If you are willing to, we would be eager to continue the discussion to better understand your thoughts.

---

### Author Rebuttal · Authors · 2024-08-07

We appreciate the reviewers' valuable time in providing constructive feedback. We have thoroughly reviewed the comments and made the necessary responses and corrections. If our responses do not fully address the reviewers' questions, we are open to further discussion.

We are honored that the reviewers have recognized the practicality and value of our task **(edWJ, Bynq)**, as well as the key contributions of this paper:

- Utilization of a visual question answering model to acquire text attribute, thereby reducing manual annotation costs **(nGMK, Bynq, beT2)**;
- Implementation of hierarchical and multi-granularity alignment **(nGMK, edWJ)**;
- Incorporation of optimal transport theory and text prompts **(nGMK, Bynq, beT2)**;
- Introduction of a novel triplet alignment loss **(nGMK, beT2)**;
- Effective validation of experimental results **(Bynq, edWJ, beT2)**.

Considering that several reviewers have expressed similar concerns about our method, we provide a comprehensive answer for all reviewers here.

**[Response to nGMK:W1, edWJ:W6, beT2:W1]: Computational complexity.**
Thanks for your valuable comment. Our OLTM achieves the trade-off between performance enhancement and computational complexity. To this end, we select several methods for comparing parameters, floating-point operations (FLOPs), and frames per second (FPS), as shown in the Table 2 below. The results show that OLTM gets remarkable performance while maintaining reasonable computational costs. The reason is that: 1) only TCA module is required for inference; 2) the VQA model is used during data processing.

*Table 2: The number of parameters, FLOPs, and FPS of different methods. VI denotes visible-infrared person re-identification.*
| **Methods** | **Field** | **Backbone** | **Reference** | **Paras(M)** | **FLOPs(G)** | **FPS** |
|-------------|-----------|--------------|---------------|--------------|--------------|---------|
| DDAG[53]    | VI        | ResNet50     | ECCV'2020     | 95.6         | 5.2      | 13.2    |
| CM-NAS[54]  | VI        | ResNet50     | ICCV'2021     | 24.5         | 5.2      | 14.8|
| CAJ[55]     | VI        | ResNet50     | ICCV'2021     | 71.6         | 5.2      | 13.1    |
| DEEN[56]    | VI        | ResNet50     | CVPR'2023     | 89.3         | 13.8         | 7.7     |
| CCSC[9]     | Sketch    | ViT          | MM'2022       | 203.9        | 383.8        | -       |
| BDG[6]      | Sketch    | ResNet50     | MM'2023       | 222.9        | 5.2      | 14.9    |
| UNIReID[7]  | Unified   | CLIP         | CVPR'2023     | 149.6        | 9.3          | 8.2     |
| OLTM (Ours) | Sketch    | CLIP         | -             | 181.9        | 6.2          |  11.7   |

**[Response to edWJ:W7, beT2:Q1]: Replaceability of VQA model.**
Thank you for bringing this issue to our attention. The VQA model is inherently substitutable. Essentially, any visual-language model which is capable of generating target attribute information from images can serve as an alternative. We also use other VQA models to demonstrate their substitutability, as shown in the Table 3 below.

*Table 3: Performance comparison of different VQA models.*
| **Methods** | **mAP** | **Rank@1** | **Rank@5** | **Rank@10** |
|-------------|---------|---------|---------|----------|
| Vilt        | 62.55   | 69.48   | 90.36   | 95.18    |
| Blip[1]     | 62.63   | 67.87   | 91.37   | 96.79    |
| GIT[2]      | 62.33   | 68.47   | 90.76   | 96.59    |


**Note: A PDF file** is attached in the common response. It contains **all the new figures** we used in the rebuttal phase.

[1]Li J, Li D, Xiong C, et al. "Blip: Bootstrapping language-image pre-training for unified vision-language understanding and generation" *ICML* 2022.

[2]Wang J, Yang Z, Hu X, et al. "Git: A generative image-to-text transformer for vision and language" *arXiv preprint* 2022.

---

### Decision · Program_Chairs · 2024-09-25

**Decision:**

Accept (poster)

**Comment:**

This paper introduces a novel framework for  Sketch Re-ID. After the author-reviewer discussion period, this paper received an average score of 5.75, with one negative score of 4. Since the reviewer with such a negative score did not engage in the discussion, I have checked the author's rebuttal carefully. I agree that the author has addressed the concerns.

Since the AC recognizes the overall quality of this paper, it is recommended to accept it to the NeruIPS'24 conference.